# Quantitative analyses reveal extracellular dynamics of Wnt ligands in *Xenopus* embryos

Yusuke Mii[1,2,3,4]*, Kenichi Nakazato[5], Chan-Gi Pack[6,7], Takafumi Ikeda[4], Yasushi Sako[6], Atsushi Mochizuki[5,8], Masanori Taira[4,9]*, Shinji Takada[1,2]*

[1]National Institute for Basic Biology and Exploratory Research Center on Life and Living Systems (ExCELLS), National Institutes of Natural Sciences, Okazaki, Japan; [2]The Graduate University for Advanced Studies (SOKENDAI), Okazaki, Japan; [3]Japan Science and Technology Agency (JST), PRESTO, Kawaguchi, Japan; [4]Department of Biological Sciences, Graduate School of Science, The University of Tokyo, Tokyo, Japan; [5]Theoretical Biology Laboratory, RIKEN, Wako, Japan; [6]Cellular Informatics Laboratory, RIKEN, Wako, Japan; [7]ASAN Institute for Life Sciences, ASAN Medical Center, University of Ulsan College of Medicine, Seoul, Republic of Korea; [8]Laboratory of Mathematical Biology, Institute for Frontier Life and Medical Sciences, Kyoto University, Kyoto, Japan; [9]Department of Biological Sciences, Faculty of Science and Engineering, Chuo University, Tokyo, Japan

*For correspondence:
mii@nibb.ac.jp (YM);
m-taira.183@g.chuo-u.ac.jp (MT);
stakada@nibb.ac.jp (ST)

**Competing interests:** The authors declare that no competing interests exist.

**Abstract** The mechanism of intercellular transport of Wnt ligands is still a matter of debate. To better understand this issue, we examined the distribution and dynamics of Wnt8 in *Xenopus* embryos. While Venus-tagged Wnt8 was found on the surfaces of cells close to Wnt-producing cells, we also detected its dispersal over distances of 15 cell diameters. A combination of fluorescence correlation spectroscopy and quantitative imaging suggested that only a small proportion of Wnt8 ligands diffuses freely, whereas most Wnt8 molecules are bound to cell surfaces. Fluorescence decay after photoconversion showed that Wnt8 ligands bound on cell surfaces decrease exponentially, suggesting a dynamic exchange of bound forms of Wnt ligands. Mathematical modeling based on this exchange recapitulates a graded distribution of bound, but not free, Wnt ligands. Based on these results, we propose that Wnt distribution in tissues is controlled by a dynamic exchange of its abundant bound and rare free populations.

## Introduction

The Wnt family of secreted signaling proteins has diverse roles in animal development, stem cell systems, and carcinogenesis (*Clevers et al., 2014*; *Loh et al., 2016*; *Nusse and Clevers, 2017*). It has been generally accepted that in the extracellular space, morphogenic Wnt ligands form a concentration gradient by dispersal (*Clevers et al., 2014*; *Kiecker and Niehrs, 2001*; *Müller et al., 2013*; *Smith, 2009*; *Strigini and Cohen, 2000*; *Tabata and Takei, 2004*; *Yan and Lin, 2009*; *Zecca et al., 1996*; *Zhu and Scott, 2004*). In contrast to this classical view, evidence also suggests dispersal-independent functions of Wnt ligands. For instance, a membrane-tethered form of Wingless (Wg) can recapitulate an almost normal pattern of *Drosophila* wings, suggesting that dispersal of Wg is dispensable for patterning (*Alexandre et al., 2014*). This dispersal-independent patterning can be explained by gradual attenuation of Wg expression in distally localized cells in which Wg was formerly expressed. However, it remains unclear to what extent dispersal-dependent and/or -independent mechanisms contribute to the graded distribution of Wnt proteins in tissue patterning.

Visualization of Wnt ligands is essential to understand their distributions. In the wing disc of *Drosophila*, Wg proteins are widely distributed from wing margin cells, where Wg is expressed (*Strigini and Cohen, 2000*; *Zecca et al., 1996*). Furthermore, long-range dispersal of Wg was evidenced by an experiment in which Wg was captured by distally expressed Frizzled2, a Wg receptor (*Chaudhary et al., 2019*). Similarly, endogenous Wnt ligands tagged with fluorescent proteins showed long-range distributions in *C. elegans* (*Pani and Goldstein, 2018*). In addition to these observations in invertebrates, we found that endogenous Wnt8 ligands disperse far from their source cells in *Xenopus* embryos (*Mii et al., 2017*). On the other hand, mouse Wnt3 accumulates within a few cell diameters of its source cells in the microenvironment of the intestine (*Farin et al., 2016*). These studies show that Wnt ligands apparently disperse in tissues and embryos, although the dispersal range varies. Importantly, in many of these studies, Wnt ligands accumulate locally on cell surfaces, showing punctate distributional patterns (*Pani and Goldstein, 2018*; *Strigini and Cohen, 2000*; *Zecca et al., 1996*). Furthermore, we demonstrated that Wnt8 and Frzb, a secreted Wnt inhibitor, accumulate separately and locally on cell surfaces in *Xenopus* embryos (*Mii et al., 2017*). However, these punctate accumulations on cell surfaces, largely ignored in the literature in the context of Wnt gradient formation, raise the question of whether such accumulations contribute to formation of concentration gradients in tissues and embryos.

Studies in *Drosophila* wing disc have shown that cell surface scaffolds, such as heparan sulfate (HS) proteoglycans (HSPGs), are required for both distribution and delivery of morphogens, including Wg, Hedgehog (Hh), and Decapentaplegic (Dpp) (*Franch-Marro et al., 2005*; *Lin, 2004*; *Yan and Lin, 2009*). From these studies, the 'restricted diffusion' model, in which morphogens are transferred extracellularly by interacting with cell surface scaffolds, has been proposed (*Yan and Lin, 2009*). In this model, the movement of each morphogen molecule is constrained in a 'bucket brigade' fashion by interactions with cell surface scaffolds. As a result of continuous interactions, morphogen molecules are slowly transferred (*Han et al., 2005*; *Yan and Lin, 2009* #152; *Kerszberg and Wolpert, 1998*; *Takei et al., 2004*). However, it seems difficult to explain local accumulations of Wnt proteins by the restricted diffusion mechanism, because passive diffusion alone should result in smoothly decreasing gradients. On the other hand, we recently showed that HSPGs on cell surfaces are discretely distributed in a punctate manner, which varies with heparan sulfate (HS) modification, forming two different types of HS clusters, *N*-sulfo-rich and *N*-acetyl-rich forms (*Mii et al., 2017*). Notably, Wnt8 and Frzb, a secreted Frizzled-related protein (sFRP), accumulate separately on *N*-sulfo-rich and *N*-acetyl-rich HS clusters, respectively. Frzb expands the distribution and signaling range of Wnt8 by forming heterocomplexes (*Mii and Taira, 2009*), and Wnt8/Frzb complexes are colocalized with *N*-acetyl-rich HS clusters (*Mii et al., 2017*). *N*-sulfo-rich clusters are frequently internalized together with Wnt8, whereas *N*-acetyl-rich HS clusters tend to remain on the cell surface. This difference in stability on the cell surface may account for the short-range distribution of Wnt8 and the long-range distribution of Frzb (*Mii and Taira, 2009*; *Mii et al., 2017*) and suggests that the distribution of HS clusters should be considered in order to understand extracellular dynamics of Wnt ligands (*Mii and Takada, 2020*).

To explain the dynamics of Wnt ligands in tissues, quantitative analyses of Wnt ligands are required. Dynamics of secreted proteins have been investigated using fluorescence recovery after photobleaching (FRAP) (*Sprague and McNally, 2005*; *Sprague et al., 2004*) and fluorescence correlation spectroscopy (FCS), although optimal ranges for diffusion coefficients differ (*Hess et al., 2002*; *Kicheva et al., 2012*; *Müller et al., 2013*; *Fradin, 2017*). For example, FRAP measurements have shown that Dpp and Wg diffuse slowly in the *Drosophila* wing disc with diffusion coefficients ranging from 0.05 to 0.10 $\mu m^2$/s, suggestive of the restricted diffusion model (*Kicheva et al., 2007*). In contrast, FCS measurements of FGF8 in zebrafish embryos showed fast, virtually free diffusion, with a diffusion coefficient of ~50 $\mu m^2$/s (*Yu et al., 2009*). Furthermore, in contrast to the FRAP results, free diffusion of Dpp measured in the *Drosophila* wing disc using FCS yielded a diffusion coefficient of ~20 $\mu m^2$/s (*Zhou et al., 2012*). FCS is based on fixed-point scanning within a confocal volume (typically sub-femtoliter) for several seconds, while FRAP evaluates considerably larger regions of photobleaching/photoconversion, containing tens or hundreds of cells (*Rogers and Schier, 2011*) and spanning long time windows (typically several hours). Under these experimental conditions for FRAP, it is proposed that diffusion of secreted proteins is affected by zigzag paths of the narrow intercellular space between polygonal epithelial cells, instead of an open, unobstructed space (hindered diffusion model) (*Müller et al., 2013*), and/or by endocytosis, which reduces the

concentration of the diffusing species in the extracellular space. Thus, we need exercise caution when comparing data derived from FRAP and from FCS analyses.

In this study, we examined extracellular dynamics of Wnt8 and Frzb, both of which are involved in anteroposterior patterning of vertebrate embryos (*Clevers and Nusse, 2012*; *Kiecker and Niehrs, 2001*; *MacDonald et al., 2009*; *Mii et al., 2017*). First, we visualized their localization in *Xenopus* embryos by fusing them with fluorescent proteins and we examined their dispersion by capturing them in distant cells. We also examined their dispersal dynamics using FCS and fluorescence decay after photoconversion (FDAP) measurements in embryonic tissue. In particular, we refined FDAP-based analysis by focusing on a limited area at the cell boundary, which enabled us to quantify dynamics comparable to those measured by FCS. Based on these results and our previous findings, we propose a basic mathematical model to explain distribution and dispersion of secreted proteins.

## Results

### Extracellular distributions of secreted proteins depend on interactions with cell-surface molecules

As we have previously shown (*Mii et al., 2017*), Wnt8 and Frzb fused with monomeric Venus (mV) were visualized along cell boundaries when expressed in *Xenopus* embryos (*Figure 1A*). We note that biological activities of these proteins were not severely impaired by the fusion of mV and that the reduced activity of mV-tagged Wnt8 compared to untagged Wnt8 could possibly be due, at least in part, to differences in translation (*Figure 1—figure supplement 1*). In contrast, we found that only the secreted form of mV (sec-mV), which was not expected to bind specifically to the cell surface, was hardly visible along the cell boundary under the same conditions (*Figure 1A*, right). Since Wnt8 and Frzb colocalize with heparan sulfate clusters on cell surfaces, we speculated that binding to cell surface proteins, like heparan sulfate proteoglycans (HSPGs), affects the distribution of Wnt8 and Frzb. To examine this possibility, we added heparin-binding (HB) peptides, consisting of 16 (ARKKAAKA)$_2$ (HB2) or 32 amino acids (ARKKAAKA)$_4$ (HB4) (*Verrecchio et al., 2000*; *Figure 1C*) to sec-mV. Addition of HB peptides significantly increased the intensity of mVenus fluorescence in the intercellular region compared to that of sec-mV. This suggests that the intercellular distribution of secreted proteins depends on interactions with docking molecules on cell surfaces.

To directly examine this idea, we constructed a reconstitution system, consisting of HA-epitope-tagged secreted mVenus (sec-mV-2HA) and a membrane-tethered anti-HA antibody ('tethered-anti-HA Ab') (*Figure 2A*, see *Figure 2—figure supplement 1* for cDNA cloning and validation of anti-HA antibody). This artificial protein and tethered-anti-HA Ab were expressed in separated areas in the animal cap region of *Xenopus* gastrulae. As with sec-mV, sec-mV-2HA was hardly visible in the intercellular space, even close to the source cells (*Figure 2B*). In contrast, sec-mV-2HA was observed around tethered-anti-HA Ab-expressing cells that were traced with memRFP, even though these cells were distantly located from the source cells (*Figure 2B*). Thus, interaction with cell surface proteins can affect distributions of secreted proteins.

This result also indicates that diffusing proteins are not readily visible using standard confocal microscopy, unless they are trapped by cell surface proteins. In fact, quantitative analysis of artificial secreted proteins revealed a slight, but significant increase of photon counts in the intercellular region by injection of mRNA for *sec-mV*, compared to uninjected embryos, indicating that sec-mV actually exists in the intercellular region (*Figure 1D*, *Figure 1—figure supplement 2*).

### Populations of secreted Wnt8 and Frzb proteins disperse long distance

We next examined dispersal of molecules of mV-Wnt8 and mV-Frzb. Both mV-Wnt8 and mV-Frzb accumulated locally along the cell boundary at the subapical level (*Figure 1A and B*), consistent with previous observations (*Mii et al., 2017*), which indicated that populations of Wnt8 and Frzb in the intercellular space were bound to the cell surface at HS clusters. On the other hand, given that some molecules of mV-Wnt8 or mV-Frzb may drift away from the cell surface, these proteins would be almost undetectable with standard confocal microscopy, as exemplified by sec-mV (*Figure 1A*) and tethered-anti-HA Ab (*Figure 2B*). To examine such mobile proteins, we tried to capture them using 'morphotrap' located distantly from the source cells (*Figure 2C*). Morphotrap is a membrane-tethered form of anti-GFP nanobody, originally devised to block dispersal of Dpp-GFP from source cells

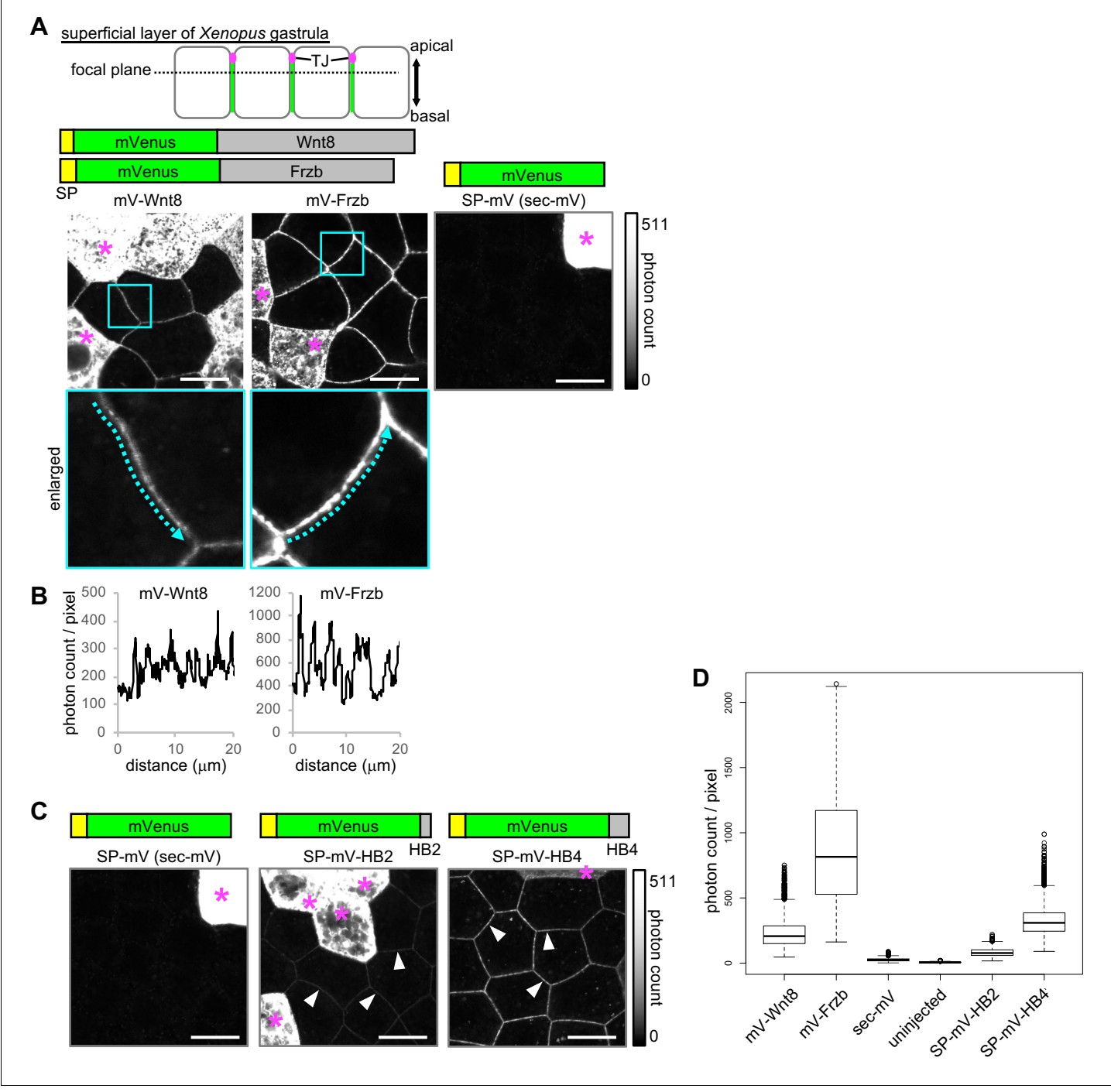

**Figure 1.** Extracellular distributions of Wnt8, Frzb, and artificial secreted proteins. All images presented were acquired using live-imaging with the photon counting method, which enables saturation-free imaging even with samples having a wide dynamic range. (**A**) Distribution of secreted proteins in the superficial layer of a living *Xenopus* gastrula (st. 10.5–11.5). Observed focal planes were at the subapical level, as illustrated. mRNAs for indicated mVenus (mV) fusion proteins were microinjected into a single ventral blastomere of four- or eight-cell stage embryos to observe regions adjacent to the source cells (indicated with asterisks). All images were acquired in the same condition with photon counting detection. Look-up tables (LUT) show the range of the photon counts in the images. (**B**) Intensity plots for mV-Wnt8 and mV-Frzb in the intercellular space. Plots along the arrows in enlarged pictures in (**A**) are shown. (**C**) Distribution of artificial secreted proteins in *Xenopus* embryos. The data of sec-mV is the same as in (**A**). sec-mV was not apparent in the intercellular space, whereas sec-mV-HB2 and sec-mV-HB4 were distributed in the intercellular space (arrowheads). SP, signal peptide; HB, heparin binding peptide. (**D**) Quantification of fluorescent intensities in the intercellular space. Photon counts per pixel are presented. All samples

*Figure 1 continued on next page*

Figure 1 continued

show statistically significant differences (p<2e-16, pairwise comparisons using the Wilcoxon rank sum test adjusted for multiple comparison with Holm's method). Scale bars, 20 μm. Amounts of injected mRNAs (ng/embryo): *mV-wnt8*, *mV-frzb*, *sp-mV*, *sp-mV-hb2*, or *sp-mV-hb4*, 0.25.

The online version of this article includes the following figure supplement(s) for figure 1:

**Figure supplement 1.** Endogenous-equivalent dose of *mV-wnt8* and biological activity of mV-Wnt8 and mV-Frzb.

**Figure supplement 2.** Imaging of secreted mVenus protein in intercellular space.

(*Harmansa et al., 2015*). We supposed that morphotraps could be utilized to detect or visualize diffusible proteins, similar to tethered-anti-HA Ab. As expected, sec-mV accumulated on the surface of morphotrap-expressing cells remote from source cells (*Figure 2C*). Similarly, mV-Wnt8 and mV-Frzb were trapped (*Figure 2C*), evidencing the long-distance dispersal (over 15 cells/200 μm) of some of secreted mV-Wnt8 and mV-Frzb molecules. These proteins are not likely to be transferred by cell-movement-based mechanisms, including distant migration of source and morphotrap-expressing cells, because cells in the animal cap region form an epithelial sheet and are tightly packed. In addition to this dispersing population, mV-Wnt8 and mV-Frzb were also detectable in gradients from producing cells to morphotrap-expressing cells, unlike the case of sec-mV (*Figure 2C and D*). These results suggest that populations of mV-tagged Wnt8 and Frzb do not associate tightly with cell surfaces, thereby potentially dispersing far from source cells.

## FCS analyses combined with quantitative imaging reveal cell-surface-bound and diffusing Wnt8 and Frzb proteins in the extracellular space

We next attempted to quantify the populations of Wnt8 or Frzb proteins associated with cell surfaces and diffusing in the extracellular space. For this purpose, we employed fluorescence correlation spectroscopy (FCS). FCS analyzes fluctuation of fluorescence by Brownian motion of fluorescent molecules in a sub-femtoliter confocal detection volume (*Figure 3*, A and B). By autocorrelation analysis (*Figure 3C*), FCS can measure diffusion coefficients (*D*) of mobile molecules and the number of particles in the detection volume, but inference of diffusion coefficients depends on mobile molecules (*Hess et al., 2002*; *Fradin, 2017*). FCS analyses were performed by injecting the same doses of mV-Wnt8 and sec-mV that were used in the experiments shown in *Figure 1* (250 pg mRNA/embryo) to consider the relationship between photon counting from live-imaging and *NoP* from FCS. Furthermore, to measure the dynamics of mV-Wnt8 and mV-Frzb at a concentration equivalent to the endogenous concentration, a decreased amount of RNA was also injected (20 pg mRNA/embryo, see *Figure 1—figure supplement 1A*).

To analyze the data obtained by FCS measurements, we compared the suitability of one-component and two-component diffusion models using the Akaike information criterion (AIC) (*Tsutsumi et al., 2016*). AIC supported fitting with the two-component model, comprising fast and slow diffusing components (*Figure 3—figure supplement 1A*). Consistent with predictions, the result indicated that the number of particles (*NoP*) of mV-Wnt8 (250 pg/embryo) was significantly higher than that of mV-Wnt8 (20 pg/embryo, endogenous-equivalent level; *Figure 3D* and *Figure 3—figure supplement 1B*). Mean values of $D_{fast}$ indicate that fast-diffusing components in all groups examined can be regarded as free diffusion (*Figure 3D*), because theoretical, as well as reported *D* values of freely diffusing proteins of similar size, range from 10 to 100 μm²/s (*Pack et al., 2006*; *Yu et al., 2009*; *Zhou et al., 2012*). Importantly, even at the endogenous-equivalent level, mV-Wnt8 and mV-Frzb show freely diffusing populations ($D_{fast}$ >10 μm²/s, *Figure 3—figure supplement 1B*). We also note that the diffusion coefficient of the fast component of mV-Wnt8 (20 pg/embryo, endogenous-equivalent level) was significantly lower than that of mV-Wnt8 (250 pg/embryo), suggesting stronger constraints with the endogenous-equivalent expression level. Thus, we conclude that within the small volume of FCS measurements, a population of mV-Wnt8 and mV-Frzb molecules diffuses freely under physiological conditions.

As shown in *Figure 1D*, photon counts of mV-Wnt8 were much higher than those of sec-mV. However, under the same conditions as in *Figure 1* (250 pg mRNA/embryo), *NoP* of mV-Wnt8 was similar to that of sec-mV (*Figure 3—figure supplement 1B*) or even smaller in another set of measurements (*Figure 3—figure supplement 2*). Thus, molecules detected in FCS appear not to contribute to the photon counts in the confocal imaging under these conditions. We speculate that FCS

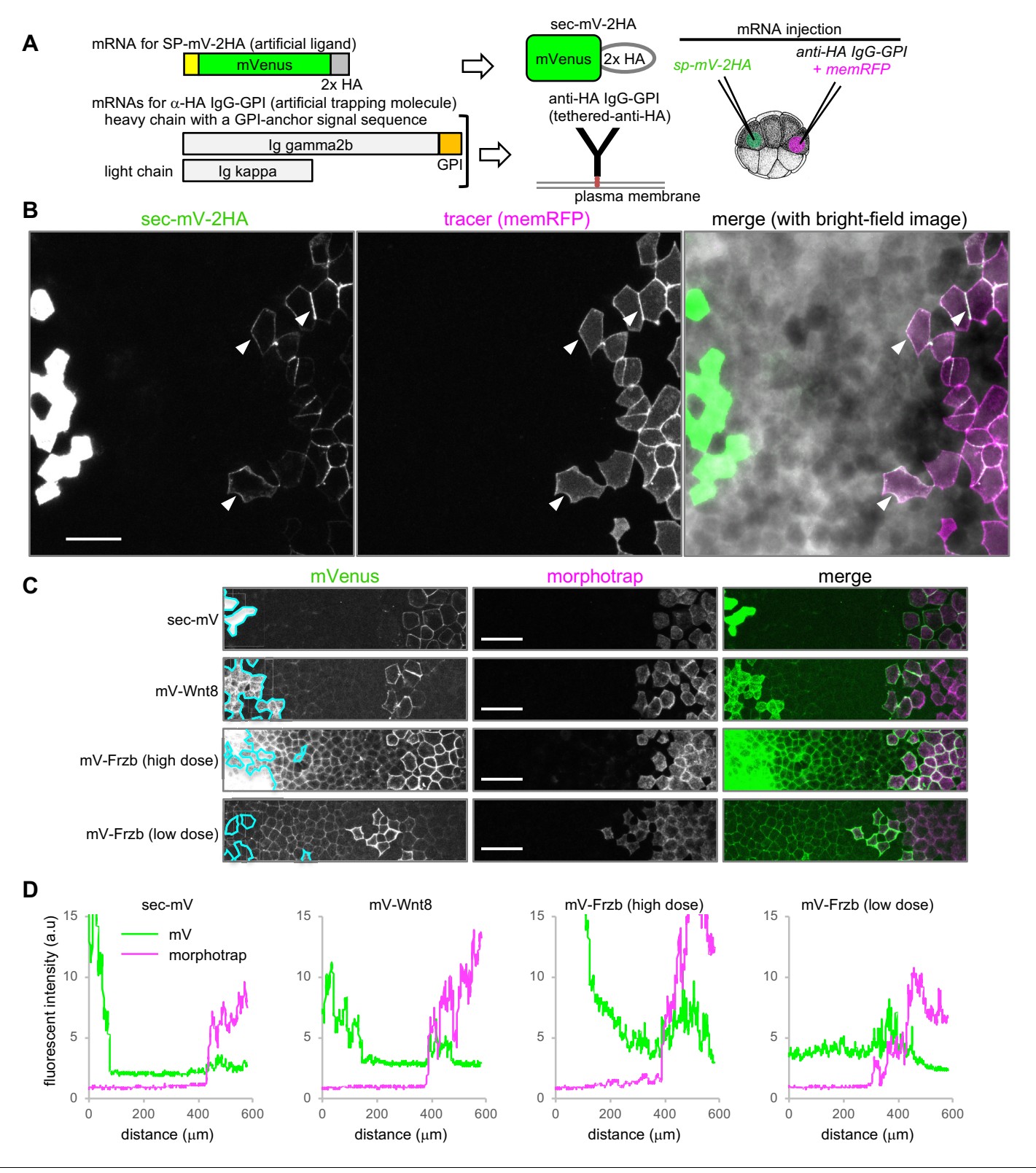

**Figure 2.** Tethered-anti-HA Ab and morphotrap. (**A**) Schematic representation of tethered-anti-HA Ab. (**B**) Results of tethered-anti-HA Ab. The artificial ligand (sec-mV-2HA) was trapped at tethered-anti-HA Ab-expressing cells, distant from the source. The superficial layer of a *Xenopus* gastrula (st. 11.5) was imaged as a z-stack and its maximum intensity projection (MIP) was presented for the fluorescent images. Intercellular mVenus signal (green) of sec-mV-2HA was not apparent in the vicinity of source cells, but was detected around the tethered-anti-HA Ab-expressing cells (arrowheads) that are

*Figure 2 continued on next page*

*Figure 2 continued*

traced with memRFP (magenta). (**C**) Morphotrap at a distant region from the source. The superficial layer of a *Xenopus* gastrula (st. 11.5) was imaged as a z-stack and its maximum intensity projection (MIP) was presented for the fluorescent images. The intercellular mVenus signal of an artificial ligand, sec-mV (green), was not detected in the vicinity of source cells (green) (left panel), but was detected around the morphotrap-expressing cells that can be traced by mCherry fluorescence (middle panels). Also, mV-Wnt8 and mV-Frzb were trapped and accumulated on distant morphotrap-expressing cells, suggesting the existence of diffusing molecules in the distant region. Source regions are indicated with cyan lines according to memBFP (tracer for mV-tagged proteins, not shown). (**D**) Distribution of mVenus and morphotrap. Fluorescent intensity of mVenus and mCherry (for morphotrap) was plotted from the left to the right. Scale bars, 100 µm. Amounts of injected mRNAs (ng/embryo) *sp-mV-2ha*, 1.0; *memRFP*, 0.15; *ig gamma2b-gpi*, 1.1; *ig kappa*, 0.63 (B); *sec-mV*, *mV-wnt8*, or *mV-frzb* (high dose), 0.25; *mV-frzb* (low dose), 0.063; *morphotrap*, 1.0; *memBFP,* 0.1 (C).

The online version of this article includes the following figure supplement(s) for figure 2:

**Figure supplement 1.** Generation of functional antibody protein by mRNA-injection into *Xenopus* embryos.

measurements might be biased to choose positions where HS-bound molecules are not abundant. Otherwise, HS-bound, immobile molecules cause strong photobleaching, which results in large drift of the fluorescence intensity. In general, such a data is not suitable for analysis.

Interestingly, slow components were observed not only in mV-Wnt8, but also in sec-mV (*Figure 3D*). To characterize these slow components, we examined the effects of HS-chain digestion with FCS. These analyses were performed with embryos injected with 250 pg/embryo RNA for mV-Wnt8 or sec-mV, because at the endogenous-equivalent level, measured values showed a large variance, possibly reflecting heterogeneity of the extracellular space, and also signal detection was difficult for sec-mV. For this purpose, we made a membrane-tethered form of Heparinase III (HepIII-HA-GPI, also known as heparitinase I) (*Hashimoto et al., 2014*). HepIII-HA-GPI enables us to digest HS chains in a region of interest (*Figure 3—figure supplement 3*), allowing us to examine the effects of HS-digestion in the same embryos. For mV-Wnt8, *NoP* and the fraction of fast components, $F_{fast}$ was significantly increased by HepIII, suggesting release of mV-Wnt8 from HS chains (*Figure 3E and F*). Thus, we suggest that HS chains contribute to the slow components of mV-Wnt8. For sec-mV, although NoP was not significantly changed by HepIII, $F_{fast}$ was slightly, but significantly increased by HepIII (*Figure 3G and H*). Furthermore, fluorescence cross-correlation spectroscopy (FCCS) analysis indicated that sec-mV did not interact with the cell membrane (*Figure 3—figure supplement 1C and D*). Thus, HS-chains showed some contribution to the slow components of sec-mV even without interaction. We speculate that such a slow population of sec-mV could be explained by hindered diffusion, in which torturous diffusion results from HS chains and other ECMs, because HS chains are highly hydrophilic and well hydrated (*Figure 3—figure supplement 1E*).

## FDAP analyses suggest exchange of cell-surface-bound and unbound states of Wnt8 and Frzb proteins

Although FCS analysis is suitable for measuring diffusing molecules, it cannot directly analyze molecules with extremely low mobility (*Hess et al., 2002*). To directly analyze dynamics of such molecules, we next employed fluorescence decay after photostimulation/photoconversion (FDAP) assays (*Matsuda et al., 2008*; *Müller et al., 2012*) in the intercellular space of *Xenopus* embryos.

Since FRAP/FDAP measurements usually examine considerably larger regions (typically containing tens or hundreds of cells) than with FCS (*Rogers and Schier, 2011*), direct comparisons of dynamics between FRAP/FDAP and FCS may need careful consideration. Therefore, we restricted the area of photoconversion to a diameter of 1.66 µm and reduced the measurement time (16 s), allowing us to obtain dynamic data in the intercellular region under conditions comparable to those for FCS (*Figure 4A*). We refer to this FDAP mode as 'cell-boundary FDAP.' In this analysis, we fused a photoconvertible fluorescent protein, mKikGR (mK) (*Habuchi et al., 2008*) to Wnt8 and Frzb (mK-Wnt8 and mK-Frzb). These fusion proteins showed distributions in embryos similar to mV-tagged proteins (*Figure 4A*), and retained biological activities (*Figure 4—figure supplement 1*). Importantly, observed distribution patterns of mK-Wnt8 and mK-Frzb were stable for up to tens of minutes (*Figure 4A*). Therefore, we assumed a steady state during the FDAP analysis (16 s).

After photoconversion, red fluorescent intensity of mK-tagged proteins was measured in the same rectangular area as photoconversion (*Figure 4B*). Because puncta of Wnt8 are often internalized with HS clusters (*Mii et al., 2017*), we excluded data in which vesicular incorporation was observed during measurement of mK-Wnt8. As an immobilized control, mK-Frzb in formaldehyde-

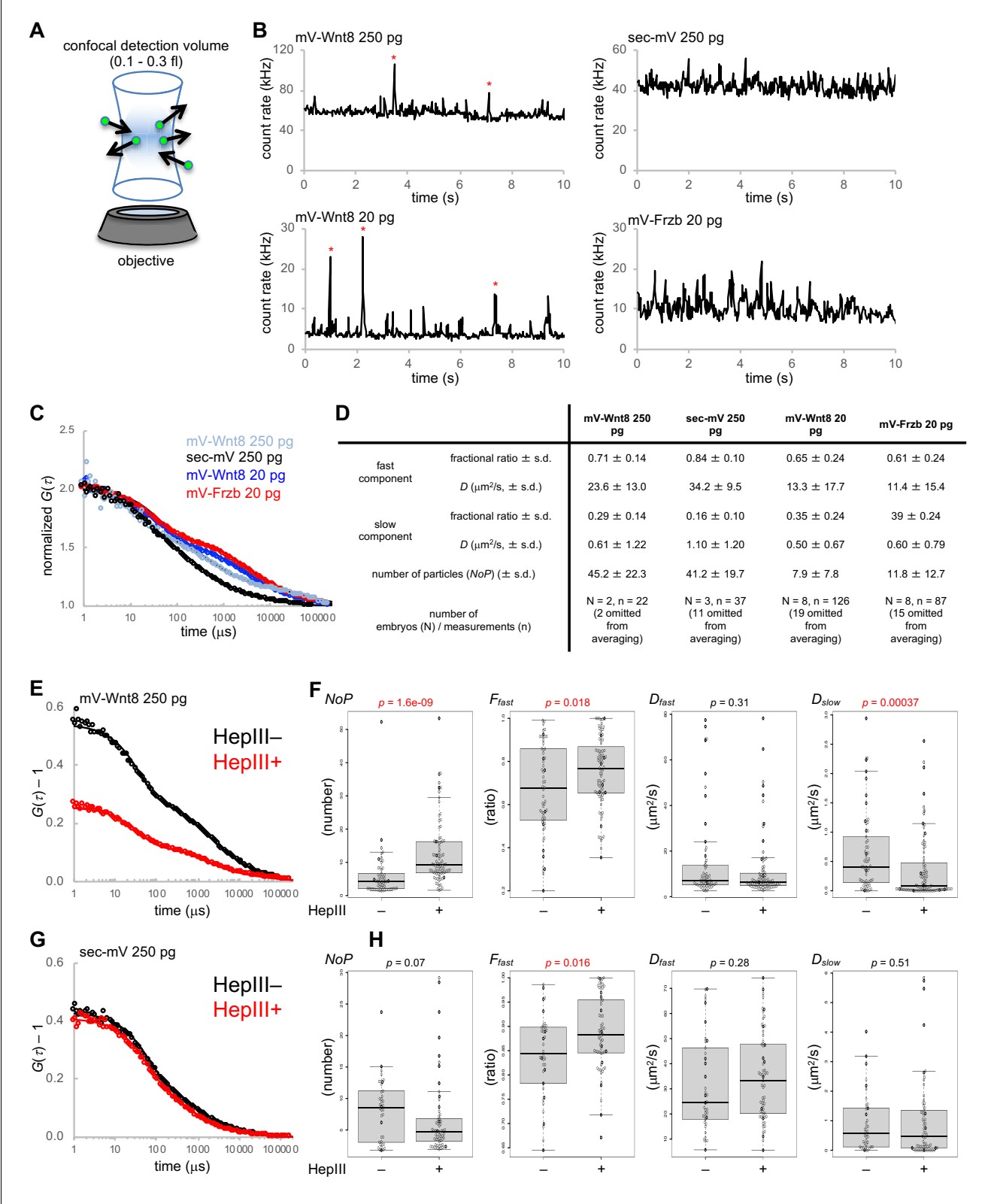

**Figure 3.** Fluorescence correlation spectroscopy (FCS) in the extracellular space of *Xenopus* embryos. mRNAs for mV-tagged proteins or sec-mV were injected into the animal pole region of a ventral blastomere of four- or eight-cell stage *Xenopus* embryo. Injected embryos were observed at gastrula stages (st. 10.5–11.5). Each FCS measurement (10 s) was performed at a point in the intercellular region within three cell diameters of the source cells. (**A**) Schematic illustration of FCS measurement. In FCS measurements, the fluorescent signal usually fluctuates due to Brownian motion of fluorescent

*Figure 3 continued on next page*

*Figure 3 continued*

molecules. Such fluctuations contain dynamic properties of fluorescent molecules. Briefly, temporal frequency of the fluctuations corresponds to the diffusion coefficient (D) and amplitude of the fluctuations corresponds inversely to the number of particles in the confocal detection volume (0.3 fl with Leica system; 0.12 fl with Zeiss system). (B) Trace of fluorescent intensities in a single measurement of indicated conditions. mV-Wnt8 shows characteristic peaks, probably corresponding to multimeric forms (asterisks, *Takada et al., 2018*). (C) Normalized autocorrelation curves of averaged data. Numbers of embryos/measurements are as indicated in (D). Experimental data are plotted with circles with the best fitting curve. (D) Summary of FCS measurements. Mean values are presented. s.d., standard deviation. Indicated numbers of measurements were omitted for averaging (in the table, no data were omitted for C), based on $D_{fast}$ values over 80 μm²/s (reflecting blinking of mVenus). (E–H) Effect of HS digestion by HepIII-GPI on mV-Wnt8 or sec-mV. Measurements were performed in the same embryos to achieve side-by-side comparison at control regions (HepIII–) or HS-digested regions (HepIII+). (E, G) Unnormalized autocorrelation curves of averaged data (number of embryos: (E) 3, (G) 4; number of measurements: (E) HepIII–, 84 HepIII+, 97; (G) HepIII–, 56 HepIII+, 87). (F) Measured parameters obtained by curve-fitting. Statistical significance (*p*, indicated in red, when significant) was calculated using the Wilcoxon rank sum test. Numbers of omitted measurements due to unreliable parameters ($D_{fast}$ values over 80 μm²/s; inadequate $F_{fast}$ values due to virtually the same $D_{fast}$ and $D_{slow}$ values): (F) HepIII–, 24;2 HepIII+, 10;3 (H) HepIII–, 17;1, HepIII+, 21;5. Lyn-mTagBFP2 and/or Lyn-miRFP703 were used to trace source cells, control regions, or HepIII +regions. Fluorescence of these tracers did not interfere with FCS measurements because these can be completely separated from mVenus. Amounts of injected mRNAs (pg/embryo): mV-wnt8, 250 or 20; mV-frzb, 20; sec-mV, 250; sp-hepIII-ha-gpi, 400; *lyn-mTagBFP2*, 100; *lyn-miRFP703*, 200.

The online version of this article includes the following figure supplement(s) for figure 3:

**Figure supplement 1.** Supplementary data for FCS analysis.
**Figure supplement 2.** FCS analysis with another system (Zeiss ConfoCor2).
**Figure supplement 3.** Membrane-tethered form of Heparinase III (HepIII-HA-GPI) digests HS chains on expressing cells.

---

fixed embryos was similarly photoconverted. Its intensity within the rectangular region was fitted to the photobleaching model using repeated laser scanning, confirming that it actually was immobilized by formaldehyde fixation (*Figure 4—figure supplement 2B*). Compared with this fixed control, mK-Wnt8 and mK-Frzb showed faster decline of fluorescent intensities (*Figure 4C* and *Figure 4—figure supplement 2A*), confirming that a population of mK-Wnt8 and mK-Frzb moved away from the photoconverted area.

Given that the punctate distribution of mK-Wnt8 and -Frzb results from their binding to HS clusters (*Mii et al., 2017*), we considered whether a simple dissociation model (*Equation 1*) is suitable for curve-fitting of FDAP data. Indeed, bleaching-corrected FDAP curves of mK-Wnt8 and mK-Frzb were well fitted to this model (*Figure 4D*; residuals were mostly within 5% and all within 10%) with the indicated parameters (*Figure 4E*, the off-rate constant $k_{off}$, and the rate of the constantly bound component *C*; for individual data plot, see *Figure 4—figure supplement 2D*). As a result, both mK-Wnt8 and mK-Frzb show large *C* values, indicating that the majority of these proteins can be considered immobile on the timescale of the measurement (*Figure 4E*). In addition, $k_{off}$ of mK-Wnt8 was significantly lower than that of mK-Frzb (*Figure 4—figure supplement 2D*), suggesting relatively rapid dissociation of mK-Frzb from the binding site. This difference appears to be consistent with FDAP spatial intensity profiles, in which photoconverted mK-Frzb, but not mK-Wnt8, accumulated in adjacent areas (*Figure 4—figure supplement 2C*, see also *Videos 1*, *2* and *3*). Thus, we conclude that most mK-Wnt8 and mK-Frzb molecules are bound, but can be exchanged with unbound molecules, and also dissociation rate values of mK-Wnt8 and mK-Frzb differ significantly.

## Mathematically modeling diffusion and distribution of secreted proteins

Based on our quantitative imaging (*Figure 1*) of tethered-anti-HA Ab and morphotrap (*Figure 2*) FCS (*Figure 3*) and FDAP (*Figure 4*), we conclude that most Wnt8 and Frzb molecules are bound to cell surfaces, while small numbers of freely diffusing molecules exist in the extracellular space. Furthermore, we have already shown that Wnt8 and Frzb utilize different types of HS clusters, *N*-sulfo-rich and *N*-acetyl-rich, as cell-surface scaffolds, respectively (*Mii et al., 2017*). Thus, we examined whether free diffusion and binding to HS clusters can explain the extracellular distribution or gradient formation of secreted proteins, using mathematical modeling.

Here, we consider two states of ligands: free and bound. The free state corresponds to the fast diffusing component in FCS, and we consider the bound component as immobile molecules. This model includes five dynamic processes: (i) ligand production, (ii) diffusion of free molecules in intercellular space, (iii) binding of ligands to HS clusters on cell surfaces, (iv) release of bound molecules

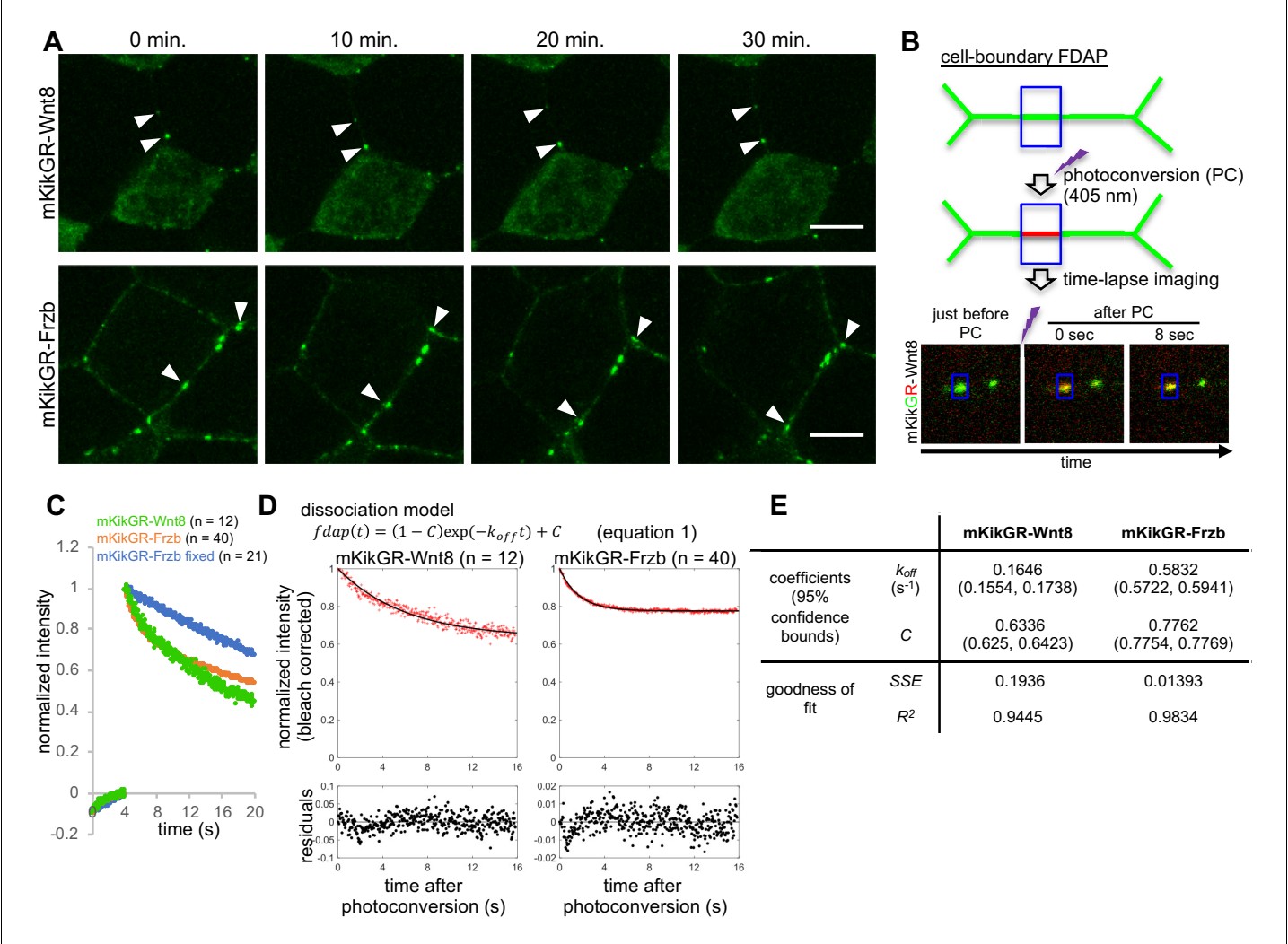

**Figure 4.** Fluorescence decay after photoconversion (FDAP) assay at the cell-boundary of *Xenopus* embryos. (**A**) Stable distribution of mKikGR-Wnt8 and mKikGR-Frzb. The superficial layer of a *Xenopus* gastrula (st. 10.5–11) was imaged as a z-stack and maximum intensity projection (MIP) was presented. Puncta of these proteins persisted for 30 min (arrowheads). Scale bars, 10 μm. (**B**) Schematic illustration of cell-boundary FDAP assay. Green lines represent mKikGR-fusion protein distributed in the intercellular region. As an example, still images before and after photoconversion (PC) are shown. Width of the blue box (area of PC and measurement) was 1.66 μm. See also *Videos 1–3* and the text for detail. (**C**) Time course of red (photoconverted state) fluorescent intensity within the photoconverted region. Photoconversion was performed about 4 s after the beginning of the measurement. Means of normalized intensities were presented (for s.d., see Figure supplement 2A). Data of 'mKikGR-Frzb fixed' were measured with MEMFA-fixed mKikGR-Frzb expressing embryos as an immobilized control. Numbers of measurements were indicated as n, which were collected in multiple experiments (twice for mKikGR-Wnt8 and mKikGR-Frzb fixed, and four times for mKikGR-Frzb). (**D**) Fluorescent decay curves fitted with the dissociation model. The mean of normalized intensities for each time point was corrected for photobleaching with division by $0.9991^{n}$ (n, number of scanning after PC; *Figure 4—figure supplement 2B*). Fitting curves are shown as black lines. Residuals were mostly within 5% (0.05) and within 10% (0.1) in all cases. (**E**) Coefficients and evaluation of goodness of fit with the dissociation model. $k_{off}$, off-rate constant; $C$, rate of immobile component; *SSE*, sum of squared errors; $R^2$, coefficient of determination. Amounts of injected mRNAs (ng/embryo): *mkikGR-wnt8* and *mkikGR-frzb, 4.0*.

The online version of this article includes the following figure supplement(s) for figure 4:

**Figure supplement 1.** Biological activity of mKikGR-Wnt8 and -Frzb.

**Figure supplement 2.** Fluorescence decay after a photoconversion (FDAP) assay in the extracellular space of *Xenopus* embryos.

**Figure supplement 3.** Fluorescent decay curve fitted with the effective diffusion model.

from HS clusters and (v) internalization of bound molecules into cells. In one-dimensional space, the model is written as:

$$\frac{\partial u}{\partial t} = D\frac{\partial^2 u}{\partial x^2} - a(x)u + bv + g(x), (0) \tag{1}$$

$$\frac{dv}{dt} = a(x)u - bv - cv, (0) \tag{2}$$

where $u$ and $v$ represent the concentration of free molecules and numbers of bound molecules, respectively, of a secreted protein. The symbols $x$ and $t$ are position and time, respectively. The symbols $a(x)$, $b$, $c$, and $g(x)$ represent binding, release, internalization, and production rates, respectively (**Figure 5A**); $a(x)$ is equivalent to the amount of HS in HS clusters (for details, see Materials and methods). $D$ (= 20 μm²/s) represents the diffusion coefficient of the free component in the extracellular space, which corresponds to the fast diffusing component measured by FCS (**Figure 3D**).

Under a wide range of appropriate parameter values, distributions of $u$ and $v$ converged to steady states within a few minutes. Compared to the fast diffusing component (**Figure 5B**), the contribution of the slow component ($D$ = 0.50 μm²/s) to the distribution range is much smaller (**Figure 5—figure supplement 1A**). Hense, we mainly consider the fast component observed in FCS (**Figure 3D**), as the diffusing population in the model. The free component, $u$, quickly decreases, displaying a shallow continuous distribution pattern due to diffusion. In contrast, the bound component, $v$, shows a discrete distribution following $a(x)$, and the level of $v$ is much higher than $u$ at any position, reflecting our conclusion that the majority of Wnt or sFRPs molecules in the extracellular space are bound. Given that activation of Wnt signaling requires internalization of the ligands (**Kikuchi et al., 2009**; **Yamamoto et al., 2006**) of the bound component, corresponding to $cv$ in **Equation 4**, the distribution of the bound component in this model could be equivalent to the 'actual' gradient of Wnt signaling, even though it is not diffusing. We demonstrated that consistently some portion of Wnt8 ligands accumulated on $N$-sulfo-rich HS clusters initiate canonical Wnt signaling by forming the signaling complex 'signalosome' (**Mii et al., 2017**).

We have shown that $N$-sulfo-rich HS, but not $N$-acetyl-rich clusters, are frequently endocytosed (**Mii et al., 2017**). In this model, different internalization rates of $N$-acetyl-rich and $N$-sulfo-rich HS clusters can be reflected by varying the internalization rate of the docking sites, $c$. A smaller value of $c$ results in long-range distribution (compare **Figure 5B and C**), explaining why Frzb shows a long-range distribution by binding to $N$-acetyl-rich HS clusters (**Mii et al., 2017**). We can evaluate the distribution by the decay length, $\lambda$. $\lambda$ represents a distribution range when the steady state gradient is written as

$$c(x) = c_o \exp(-x/\lambda) \tag{5}$$

(**Kicheva et al., 2012**; **Kicheva et al., 2007**). We calculated $\lambda$ by curve-fitting the peak values of $v$ to **Equation 5**. The value of $\lambda$ with $c$ = 0.1 or 0.01 is 6.346 or 10.79 μm (**Figure 5B,C** and **Figure 5—figure supplement 1B,C** for normalized plots), respectively, showing that internalization rates of HS clusters can affect distribution ranges, as observed between Wnt and sFRPs (**Mii and Taira, 2009**). In addition, we examined the contribution of dissociation from the bound to the diffusing state, suggested by our cell-boundary FDAP (**Figure 4**). Without dissociation, a shorter range distribution of the bound component was obtained ($\lambda$ = 4.504 μm, **Figure 5—figure supplement 1D**) than with dissociation ($\lambda$ = 6.346 μm, same data as in **Figure 5B**). Furthermore, in *Xenopus* embryos, Wnt8 in the

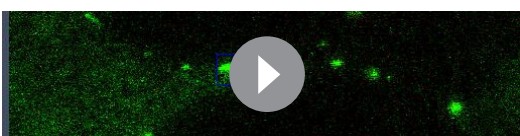

**Video 1.** Photoconversion of mKikGR-Wnt8 in a cell-boundary region of a *Xenopus* embryo.
https://elifesciences.org/articles/55108#video1

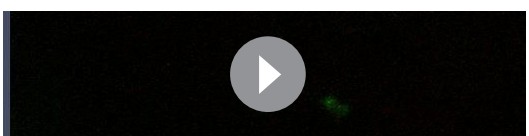

**Video 2.** Photoconversion of mKikGR-Wnt8 in a cell-boundary region of a *Xenopus* embryo (another example).
https://elifesciences.org/articles/55108#video2

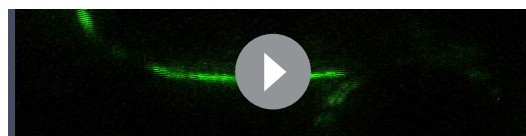

**Video 3.** Photoconversion of mKikGR-Frzb in a cell-boundary region of a *Xenopus* embryo. Photoconversion of mKikGR fusion proteins was performed at a cell-boundary region in the animal cap of a *Xenopus* gastrula (st. 10.5 st.11.5). mKikGR-Wnt8 (*Videos 1* and *2*) or mKikGR-Frzb (*Video 3*) was photoconverted at the region indicated with the blue box after 100 frames scanned (about 4 s), and another 400 frames were scanned for measurement. The width of the region for photoconversion and intensity measurement was 20 pixels (1.66 μm). The play speed is ×1.

https://elifesciences.org/articles/55108#video3

intercellular space exhibited local accumulations (*Figure 1B*). In our model, when the binding rate $a_{n,max}$ (*Equation 7* in the Materials and methods) fluctuates randomly (i.e. the amount of HS at position $x$), the bound ligand component also fluctuates (*Figure 5D*, blue), reproducing the local accumulation of Wnt8 and Frzb in *Xenopus* embryos. Even under these conditions, the free component shows a continuously decreasing gradient (*Figure 5D*, red), which probably corresponds to the FCS-measured, diffusing component of the FGF8 gradient in zebrafish embryos (*Yu et al., 2009*; measuring concentrations by FCS in a wide field is technically difficult in larger *Xenopus* embryos). Thus, our mathematical model can generalize protein distributions in the extracellular space.

## Discussion

As one of the major secreted signaling molecules, mechanisms of Wnt dispersal are crucial when we consider embryonic patterning and various other systems involving Wnt signaling (*Routledge and Scholpp, 2019*). Among many Wnt proteins, Wg distribution in the *Drosophila* wing disc has long been investigated as a morphogen gradient (*Strigini and Cohen, 2000*; *Zecca et al., 1996*). Various genetic studies show that the extracellular distribution of Wg largely depends on HSPGs, such as Dally and Dally-like glypicans (*Baeg et al., 2004*; *Franch-Marro et al., 2005*; *Han et al., 2005*). Furthermore, FRAP-based analysis suggests that the effective diffusion coefficient of Wg is much slower (0.05 μm²/s) than free diffusion (>10 μm²/s) (*Kicheva et al., 2007*). However, such dynamics of secreted signaling proteins still remain a matter of debate (*Rogers and Schier, 2011*). On the other hand, recently we found that HS chains on the cell surface are organized in clusters with varying degrees of *N*-sulfo modification in *Xenopus* embryos and HeLa cells. Furthermore, we demonstrated that endogenous Wnt8 protein visualized by immunostaining shows a punctate distribution, specifically associated with *N*-sulfo-rich HS clusters (*Mii et al., 2017*). Similar punctate distributions have also been observed with Wg in *Drosophila* (*Strigini and Cohen, 2000*; *van den Heuvel et al., 1989*), but the significance of these distributions has not yet been explained. Therefore, to gain insight into the mechanism of Wnt distribution, we examined Wnt8 protein dynamics.

Based on quantitative live-imaging techniques, we propose that most Wnt8 molecules distributed among cells are mostly cell-surface-bound, while a small portion of them are diffusing. Similarly, Wnt/EGL-20 shows that puncta mostly overlap with Frizzled and a small population of mobile/diffusing molecules is also suggested in *C. elegans* (*Pani and Goldstein, 2018*). In *Xenopus* embryos, Frizzled may also contribute to bind Wnt ligands because some Wnt8 puncta overlapped with Frizzled8 (*Mii et al., 2017*). Furthermore, a small population of diffusing Dpp has been shown in *Drosophila* wing disc (*Zhou et al., 2012*). Importantly, it has been suggested that these populations disperse over long distances, similar to our observation of mV-Wnt8 trapped using morphotrap (*Figure 2C, D*), generalizing the existence of long-dispersing populations in various model systems.

It is plausible that cell-surface-bound Wnt8 is mostly associated with HS clusters (*Mii et al., 2017*). The function of HSPGs in Wnt dispersal has been examined by genetic studies of *Drosophila*. These studies show that HSPGs are required for accumulation and transfer of Wnt ligands. Based on these results, it has been proposed that Wnt disperses by restricted diffusion, in which HSPGs transfer Wnt ligands in a bucket brigade manner (*Yan and Lin, 2009*). In our FDAP assay, most photoconverted mK-Wnt8 does not diffuse laterally, even when other puncta of Wnt8 exist near the site of photoconversion (*Figure 4—figure supplement 2C*, left panel, *Video 1*). We further considered this observation with modeling (*Figure 5—figure supplement 1E*). Unlike the experiment, modeling shows lateral dispersal of photoconverted molecules in neighboring regions, over time. For this difference, we mainly consider two possibilities: (i) Our imaging system may not be sufficiently sensitive to detect such a small increase, or the increase of the ligands may be obliterated by photobleaching.

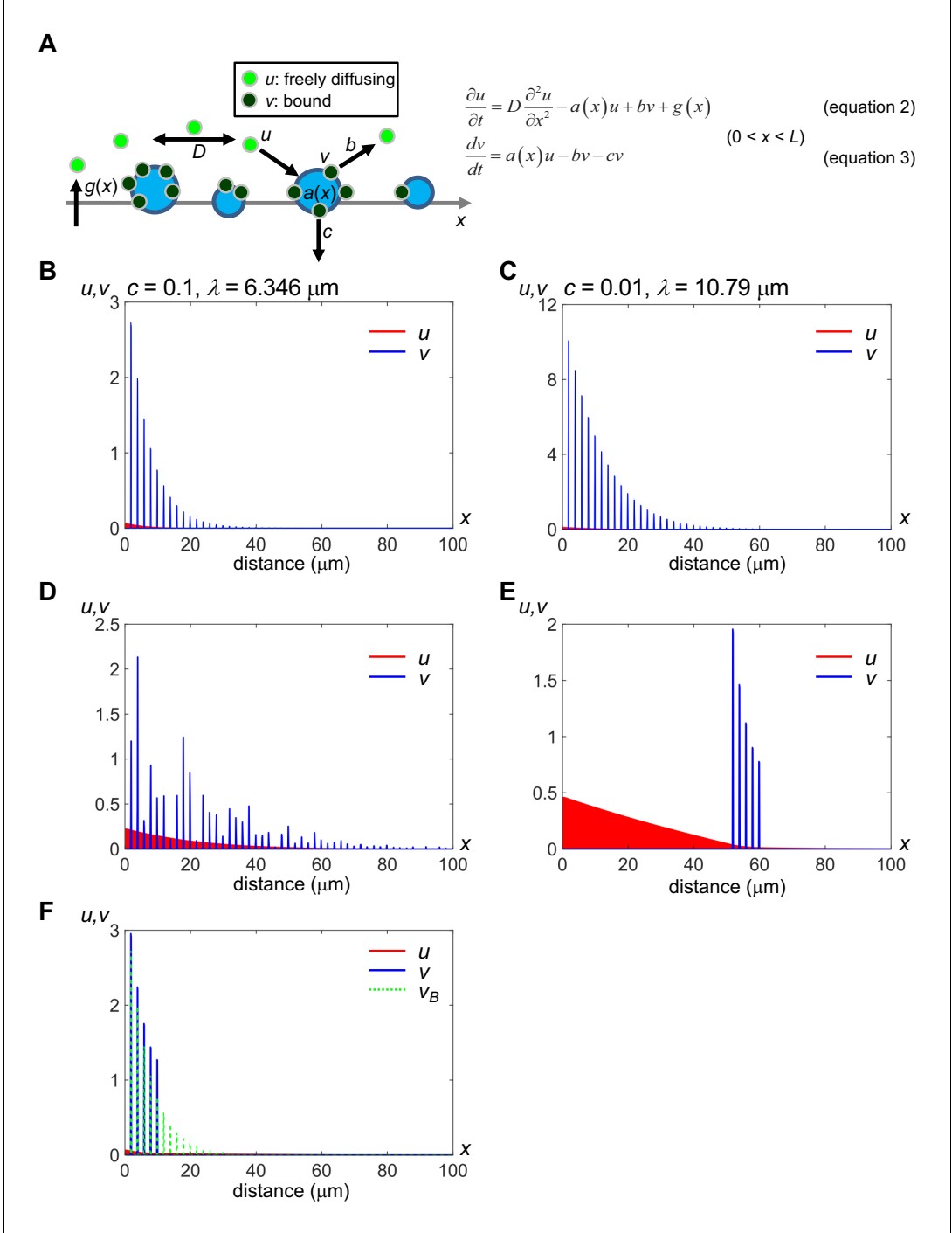

**Figure 5.** A minimal model of secreted protein dynamics in the extracellular space. Distributions of free ($u$) and bound ($v$) components of secreted proteins were obtained by computer simulation. The vertical axis indicates the amount of $u$ and $v$, and the horizontal axis indicates the distance ($x$); Distributions in the range of $0 \leq x \leq 100$ (μm) are shown while the model considers a field whose spatial length $L = 1000$ (μm). Distributions of $u$ (red) and bound $v$ (blue) at time $t = 100$ (sec) are shown, which we confirmed as being nearly steady states. We used the forward difference method with spatial step $\Delta x = 0.1$ and temporal step $\Delta t = 0.0001$ in numerical calculations. The level of $v$ at the position where docking sites exist ($a(x) = a_{n,max}$) remains relatively high even after $a_{n,max}$ exceeded $b$. (**A**) Schema of the modeling. $a(x)$, binding rate at position $x$. Note that $a(x)$ is equivalent to the amount of HS for an HS cluster. $b$, release rate from the HS clusters. $c$, internalization rate of the HS clusters. $D$, diffusion coefficient of $u$. $g(x)$, production rate at position $x$. For details, see Materials and methods. (**B**) Rapid internalization of the docking sites. Parameter values are: $D = 20.0$ (μm²/s), $a_{n,max} = 10.0$, $b = 0.1$, $c = 0.1$, $g_{max} = 0.2$, $R = L/1000$, $p_1 = 2$, and $p_2 = 0.2$. The decay length $\lambda$ is calculated as 6.346 μm, according to the fitting curve (see *Figure 5—figure supplement 1A*). See also *Source code 1*. (**C**) Slow internalization of the docking sites. Parameter values are the same as in (**A**) except for $c = 0.01$. The decay length $\lambda$ is calculated as 10.79 μm, according to the fitting curve (see *Figure 5—figure supplement 1B*). *Figure 5 continued on next page*

Figure 5 continued

This value represents a wider range than that in (**A**). See also **Source code 2**. (**D**) Local accumulation similar to intercellular distribution of mV-Wnt8 and -Frzb. $a_{n,max}$ is given randomly for each $n$ by an absolute value of the normal distribution. See also **Source code 3**. (**E**) Distant scaffolds from the source region. $a_{n,max}$ is given to depend on space: 10.0 for $50 \leq x \leq 60$, otherwise, $a_{n,max}$ is 0.0. This situation is similar to tethered-anti-HA Ab (**Figure 2B**). See also **Source code 4**. (**F**) Ligand accumulation in front of the HS-absent region. $a_{n,max}$ is given depending on space: 10.0 for $0 \leq x \leq 10$ and 0 for $10 < x \leq 1000$. Values of $v$ in (**B**) ($v_B$) are also shown with green dashed lines, for comparison. Note that ligand accumulation occurs in front of the HS-absent region ($10 < x$.). See also **Source code 5**.

The online version of this article includes the following figure supplement(s) for figure 5:

**Figure supplement 1.** Distribution ranges of the ligands in some conditions.

(ii) If binding dynamics of mK-Wnt8 are slower than in our model, ligands may diffuse away before re-binding in neighboring regions. On the other hand, mK-Frzb showed some lateral dispersal similar to the model. As previously discussed, these behaviors in FRAP experiments can be classified into some cases including 'reaction dominant' and 'effective diffusion' by a balance among the on-rate, the off-rate, and the diffusion coefficient (*Sprague and McNally, 2005*; *Sprague et al., 2004*). Restricted diffusion can be understood as a kind of effective diffusion in which dynamics of ligand binding/dissociation to HSPGs are similar to those of free diffusion. Although we did not derive binding constants, at least superficially, mK-Frzb showed an effective diffusion-like behavior, whereas mK-Wnt8 showed a reaction dominant-like behavior, in which free diffusion is much faster than binding/dissociation. In order to compare our data with those previously reported (*Kicheva et al., 2012*), we also performed curve-fitting with an effective/apparent diffusion model (*Figure 4—figure supplement 3*, *Equation 2*). As a result, the apparent diffusion coefficient $D_a$ (μm$^2$/s) was calculated as 0.042 and 0.059 for mKikGR-Wnt8 and mKikGR-Frzb, respectively. These values are very close to a previously reported FRAP value for Wg (0.05 μm$^2$/s) (*Kicheva et al., 2007*). Thus, such small values of $D_a$ relative to free diffusion could be interpreted as the result of interaction with cell surfaces, regardless of whether the protein of interest actually shows lateral diffusion in bucket brigade fashion.

We found that sec-mV is almost invisible with standard confocal microscopy (*Figure 1B*). Furthermore, binding to cell surface molecules such as HSPGs and membrane-tethered antibody was sufficient for visible distribution for artificial secreted proteins (*Figures 1C* and *2B*). These findings are similar to recent demonstrations that secreted GFP can be synthetic morphogens with specific scaffold molecules in the *Drosophila* wing disc (*Stapornwongkul et al., 2020*) and in cultured cells (*Toda et al., 2020*). On the other hand, secreted GFP appears visible in some tissues, such as deep cells in early zebrafish embryos (*Yu et al., 2009*) and developing zebrafish brain (*Veerapathiran et al., 2020*). In the zebrafish brain, secreted EGFP did not show slow components (*Veerapathiran et al., 2020*), which is different from our observation in the *Xenopus* animal cap region (*Figure 3D*). Together, considering detection of diffusing molecules (Appendix), we speculate that these differences may reflect narrowness of the extracellular space in the tissues.

Our cell-boundary FDAP suggests that cell-surface-bound and diffusing populations are probably exchangeable. Although this result can be explained by dissociation of molecules from the bound state as described above, it also seems possible that endocytosis reduces the number of photoconverted molecules (*Figure 4C*). However, we consider this less likely. Endocytosis of Wnt8 is possibly mediated by caveolin (*Mii et al., 2017*; *Yamamoto et al., 2006*), and we have already shown that internalized Wnt8 was detected as puncta in the cell (*Mii et al., 2017*). However, in FDAP analyses in this study, we excluded observations with internalization of Wnt puncta from curve-fitting analysis. In our mathematical model, when dissociation from the cell-surface does not occur ($b = 0$ in *Equation 3 and 4*, *Figure 5A*), the range of the gradient (decay length, $\lambda$) was shortened from 6.35 to 4.50 μm (*Figure 5—figure supplement 1D*). Thus, at least in cases we analyzed, dissociation from the bound state seems to contribute to the long-range distribution and rapid formation of the gradient.

A goal of this study is to link quantitative measurements of local protein dynamics to larger spatiotemporal patterns of extracellular protein dispersal in embryos. We hypothesized that local dynamics of diffusion and interaction with HS chains measured by FCS and FDAP could be extrapolated to explain mechanisms for gradient formation across many cells. We mainly consider protein dispersal within a single plain, and this is exemplified in the animal cap region since mV-Wnt8 and

mV-Frzb accumulated on the proximal side (to the source) of morphotrap-expressing cells (*Figure 2C*). But when we consider dispersal of secreted proteins in embryos, other routes can be involved. For example, a BMP antagonist, Chordin exhibits dispersal within the Brachet cleft, which is a fibronectin-rich ECM (*Plouhinec et al., 2013*). In addition, several other mechanisms, such as cell lineage-based dilution (*Farin et al., 2016*) and cytonemes/signaling filopodia (*Roy et al., 2011*; *Stanganello et al., 2015*) may contribute to dispersal of a morphogen. We emphasize that immobilization of morphogen molecules is a prerequisite for cytoneme/filopodium-mediated transfer of signaling. Gradient formation over long ranges has not been examined experimentally in this study. However, we attempted to understand the outcome of diffusion and binding, basic properties of morphogens. Thus, we propose a mathematical model consisting of free and bound components of Wnt based on observed local dynamics (*Figure 5A*). This model can be widely applied to secreted proteins that bind to cell surfaces, including sFRPs and other peptide growth factors. Notably, in our mathematical model, distributions of both free and bound components converged to steady states within a few minutes, showing rather fast dynamics in the context of embryonic patterning. This characteristic could solve perceived weaknesses of diffusion-base models (*Müller et al., 2013*), especially dilemmas related to the speed and stability of gradient formation. From this point of view, the combination of abundant cell-surface-bound and minimal diffusing populations would be beneficial for signaling stability and speed of pattern formation, respectively. Like tethered-anti-HA Ab (*Figure 2B*), atypical distributions of FGF (*Shimokawa et al., 2011*) and Nodal (*Marjoram and Wright, 2011*), in which ligands accumulate in locations distant from their sources, have been reported, although a theoretical explanation of these atypical distributions has proven elusive. In our model, atypical distributions can be reproduced if specific scaffolds for ligands (ligand binding proteins) are anchored on the surfaces of cells (*Figure 5E*). Furthermore, our model explains the puzzling localization of ligands in tissues. In mosaic analyses of the wing discs of *Drosophila* mutants, Hh and Dpp ligands accumulate at the edges of clones defective in HS synthesis (*Takei et al., 2004*; *Yan and Lin, 2009*). Distributional patterns of these ligands are explained by our model, which accounts for accumulations of ligand in regions lacking HS (*Figure 5F*). Thus, our model provides a basic framework to understand of the extracellular behavior of secreted proteins.

# Materials and methods

## Key resources table

| Reagent type (species) or resource | Designation | Source or reference | Identifiers | Additional information |
|---|---|---|---|---|
| Antibody | HepSS-1 (mouse monoclonal, IgM) | *Kure and Yoshie, 1986* *Mii et al., 2017* | | 1:400 *Figure 3—figure supplement 3C* |
| Antibody | NAH46 (mouse monoclonal, IgM) | *Suzuki et al., 2008* *Mii et al., 2017* | | 1:50 *Figure 3—figure supplement 3C* |
| Antibody | F69-3G10 (mouse monoclonal, IgG2b) | Seikagaku Corp. | 370260 | 1:200 *Figure 3—figure supplement 3AB* |
| Antibody | Anti-HA (rabbit polyclonal) | MBL | #561 | 1:200 *Figure 3—figure supplement 3ABC* |
| Antibody | Anti-Wnt8 (rabbit antiserum) | *Mii et al., 2017* | | 1:4000 *Figure 1—figure supplement 1A* |
| Antibody | Anti-mouse IgG-AlexaFluor 488 (goat polyclonal) | Invitrogen | A11029 | 1:500 *Figure 3—figure supplement 3AB* |
| Antibody | Anti-rabbit IgG-AlexaFluor 555 (goat polyclonal) | Invitrogen | A21434 | 1:500 *Figure 3—figure supplement 3AB* |
| Antibody | Anti-rabbit IgG-AlexaFluor 568 (goat polyclonal) | Invitrogen | A11011 | 1:500 *Figure 3—figure supplement 3C* |

*Continued on next page*

*Continued*

| Reagent type (species) or resource | Designation | Source or reference | Identifiers | Additional information |
|---|---|---|---|---|
| Antibody | Anti-mouse IgM-AlexaFluor 488 (goat polyclonal) | Invitrogen | A21042 | 1:500 *Figure 3—figure supplement 3C* |
| Antibody | Anti-rabbit IgG-AlexaFluor 647 (donkey polyclonal) | Invitrogen | A21245 | 1:500 *Figure 1—figure supplement 1A* |
| Antibody | Anti-mouse IgM-AlexaFluor 488 (goat polyclonal) | Invitrogen | A21042 | 1:500 *Figure 3—figure supplement 3C* |
| Cell line (*Mus musculus*) | Hybridoma anti-HA (clone 12CA5) | *Field et al., 1988* | | Mouse monoclonal, IgG2b, kappa |
| Cell line (*Mus musculus*) | Hybridoma anti-Myc (clone 9E10) | *Evan et al., 1985* | | Mouse monoclonal, IgG1, kappa |
| Gene (*Mus musculus*) | 12CA5-ig-gamma-2b | This study | LC522514 | Gene |
| Gene (*Mus musculus*) | 12CA5-ig-kappa | This study | LC522515 | Gene |
| Recombinant DNA reagent | morphotrap | *Harmansa et al., 2015* | | |
| Recombinant DNA reagent | pET21b-Phep_3797 (plasmid) | *Hashimoto et al., 2014* | | |
| Recombinant DNA reagent | pCSf107-SP-HepIII-HA-GPI (plasmid) | This study | | |
| Software, algorithm | PyCorrFit | *Müller et al., 2014* | Version 1.1.7 | Windows version |
| Software, algorithm | Fiji | *Schindelin et al., 2012* | | |
| Software, algorithm | image J | NIH | | |
| Software, algorithm | Zen2009 | Zeiss | | |
| Software, algorithm | Matlab Curve Fitting Toolbox | Mathworks | | |
| Software, algorithm | R | The R Foundation | | |

All experiments using *Xenopus laevis* were approved by the Institutional Animal Care and Use Committee, National Institutes of Natural Sciences (Permit Number 18A038, 19A062, 20A053), or the Office for Life Science Research Ethics and Safety, University of Tokyo.

## *Xenopus* embryo manipulation and microinjection

Unfertilized eggs of *Xenopus laevis* were obtained by injection of gonadotropin (ASKA Pharmaceutical). These eggs were artificially fertilized with testis homogenates and dejellied using 4% L-cysteine (adjusted to pH 7.8 with NaOH). Embryos were incubated in 1/10x Steinberg's solution at 14–17°C and were staged according to *Nieuwkoop and Faber, 1967*. Synthesized mRNAs were microinjected into early (2–16 cell) embryos. Amounts of injected mRNAs are described in figure legends.

## Fluorescent image acquisition

Image acquisition was performed using confocal microscopes (TSC SP8 system with HC PL APO ×10/NA0.40 dry objective or HC PL APO2 ×40/NA1.10 W CORR water immersion objective, Leica or LSM710 system with C-Apochromat 40x/1.2 W Corr M27 water immersion objective, Zeiss). Photon counting images were acquired with a HyD detector (Leica). Detailed conditions for imaging are available upon request. mV was constructed by introducing an A206K mutation to prevent protein aggregation (*Zacharias et al., 2002*). For FDAP and FCS measurements, gastrula embryos were embedded on 35 mm glass-based dishes (Iwaki) with 1.5% LMP agarose (#16520–050; Invitrogen)

gel, which was made of 1/10x Steinberg's solution. For other types of live-imaging, embryos were mounted in a silicone chamber made in-house with holes 1.8 mm in diameter. Fluorescent intensity was measured using Fiji, Image J (NIH) or Zen2009 (Zeiss).

## Cell lines

Hybridomas (derived from mouse, 12CA5, anti-HA; 9E10, anti-Myc) were used to obtain their total RNA and subsequent cloning of immunoglobulin genes. These hybridomas have been neither authenticated nor tested for mycoplasma because no assays were performed with these hybridomas themselves. Instead, we confirmed generation of functional anti-HA or anti-Myc IgG from the cloned genes by co-IP assay.

## Immunostaining of *Xenopus* embryos

Immunostaining of *Xenopus* embryos was carried out according to a previous report (*Mii et al., 2017*). Briefly, embryos were fixed with MEMFA (0.1 M MOPS pH 7.4, 4 mM EGTA, 2 mM MgSO$_4$, 3.7% formaldehyde) 2 hr at room temperature. Fixed embryos were dehydrated with EtOH (EtOH treatment improves staining with anti-Wnt8 and anti-HS antibodies). After rehydration, embryos were washed with TBT (1x TBS, 0.2% BSA, 0.1% Triton X-100) and blocked with TBTS (TBT supplemented with 10% heat-treated [70℃, 40 min] fetal bovine serum). The following procedures are similar for primary and secondary antibodies. Antibody was diluted with TBTS and was centrifuged 15 min at 15,000 rpm before use. Embryos were incubated with the supernatant of antibody solution overnight at 4℃. Then embryos were washed five times with TBT.

## cDNA cloning of IgG from cultured hybridomas

Cultured hybridomas were harvested by centrifugation and total RNAs were prepared using ISO-GEN (Nippon Gene), according to the manufacturer's protocol. First strand cDNA pools were synthesized using SuperScript II reverse transcriptase (Invitrogen) and random hexamer oligo DNA. These cDNA pools were used as templates for PCR to isolate cDNAs for heavy chains and light chains of anti-HA and anti-Myc IgGs. See *Supplementary file 1* for all primers used for PCR cloning. Full-length cDNAs were cloned into the pCSf107mT vector (*Mii and Taira, 2009*).

Cultured hybridoma cells were harvested by centrifugation and total RNAs were prepared using ISOGEN (Nippon Gene), according to the manufacturer's protocol. First strand cDNA pools were synthesized using SuperScript II reverse transcriptase (Invitrogen) and random hexamer oligo DNA. These cDNA pools were used as templates for PCR to isolate cDNAs for heavy and light chains of anti-HA and anti-Myc IgGs.

Procedures for PCR cloning were as follows. IgG cDNAs for 3' regions of CDSs were obtained by PCR with degenerate primers (5' γ-F and 5' κ-F) and primers corresponding to constant regions of Ig genes (γ2b-const-R, γ1-const-R, 3' κ-R) (*Wang et al., 2000*). To obtain the complete CDSs, 5'RACE was carried out to obtain the first codons of Ig genes, using a modified protocol in which inosines are introduced into the G-stretch of the HSPst-G10 anchor (personal communications from Dr. Min K. Park). cDNAs were synthesized with gene-specific primers (HA-heavy-R1, Myc-heavy-R1, HA-light-R1, and Myc-light-R1), and tailed with poly-(C) by terminal deoxynucleotidyl transferase, and subsequently double-stranded cDNAs were synthesized with the HSPst-G10 anchor. 5' ends of cDNAs were amplified by PCR between the HSPst adaptor and gene-specific primers (HA-heavy-R2, Myc-heavy-R2, HA-light-R2 and Myc-light-R2) using the double-stranded cDNAs as templates. Full length CDSs were amplified using primers designed for both ends of the CDSs (HH-Bam-F, MH-Bam-F, HL-bam-F, and ML-Bgl-F for 5' ends; 3' γ2b-R, 3' γ1-R and 3' κ-R for the 3' end) and the first cDNA pools. See *Supplementary file 1* for all primers used for PCR cloning. Full-length cDNAs were cloned into the pCSf107mT vector (*Mii and Taira, 2009*). Sequence data for anti-HA IgG genes have been deposited in Genbank/DDBJ under accession codes LC522514 and LC522515.

## FDAP measurements

For expression in the animal cap region of *Xenopus* embryos, four-cell-stage *Xenopus laevis* embryos were microinjected with mRNAs for mK-Wnt8 and mK-Frzb (4.0 ng/embryo) at a ventral blastomere. Injected embryos were incubated at 14℃ until the gastrula stage (st. 10.25–11.5) for subsequent confocal analysis. FDAP measurements were performed using the LSM710 system (Zeiss) with a

C-Apochromat ×40, NA1.2 water immersion objective. Time-lapse image acquisition was carried out for 20 s each at 25 frames/s, and after 4 s (100 frames) from the start, intercellular mK-fusion proteins were photoconverted at a small rectangular region (1.66 × 2.49 µm) with 405 nm laser irradiation. After photoconversion, images were acquired for 16 s (400 frames). Red fluorescent intensities within the rectangular region where photoconversion was performed, were analysed by curve-fitting to *Equation 1* (*Figure 4D*) or *Equation 2* (*Figure 4—figure supplement 3A*), using the Curve Fitting Toolbox of MATLAB (Mathworks).

## FCS measurements

FCS measurements were carried out using a ConfoCor2 system (objective: C-Apochromat ×40, NA1.2 water immersion) (Zeiss; *Figure 3—figure supplement 2* only) according to a previous report (*Pack et al., 2006*) or a TSC SP8 equipped with FCS (objective: HC PL APO 63x/1.20 W motCORR CS2) (Leica). mRNAs for mV-Wnt8 and sec-mV were microinjected into four- or eight-cell stage *Xenopus* embryos. Injected embryos were measured at gastrula stage (st. 10.5–11.5). Rhodamine 6G (Sigma-Aldrich) was used to calibrate detection volume, with a reported value of its diffusion coefficient (280 µm$^2$/s) (*Pack et al., 2006*). PyCorrFit software (*Müller et al., 2014*) was used for curve-fitting analyses of FCS data from the Leica system. Models considering three-dimensional free diffusion with a Gaussian laser profile, including a triplet component ('T + 3D', a one-component model or 'T + 3D + 3D', a two-component model) were used for fitting. Akaike information criterion (AIC) was used to compare fitting with the one-component and two-component models according to a previous report (*Tsutsumi et al., 2016*).

## Plasmid construction

pCSf107mT (*Mii and Taira, 2009*) was used to make most plasmid constructs for mRNA synthesis. pCSf107SPw-mT and pCSf107SPf-mT were constructed, which have the original signal peptides of Wnt8 and Frzb, respectively. The coding sequence (CDS) for mVenus (mV) or mKikGR (mK) was inserted into the BamHI site of pCSf107SPw-mT or pCSf107SPf-mT to construct pCSf107SPw-mV-mT, pCSf107SPf-mV-mT, pCSf107SPw-mK-mT, and pCSf107SPf-mK-mT. Constructs for SP-mV, SP-mV-HB, and SP-mV-2HA were made with pCSf107SPf-mT. pCS2 +HA-IgH-TM-2FT (the heavy chain for anti-HA IgG with the transmembrane domain of a membrane-bound form of IgG heavy chain) was made by inserting the full length CDS of heavy chain of anti-HA IgG without the stop codon (using the EcoRI and BglII sites) and a partial CDS fragment corresponding to the IgG transmembrane domain (using the BglII and XbaI sites) into the EcoRI/XbaI sites of pCS2 +mcs-2FT-T. To construct pCSf107-SP-HepIII-HA-GPI, HepIII CDS was inserted into pCSf107-SP-mcs-4xHA-GPI.

## Luciferase reporter assays

Luciferase reporter assays were carried out as previously described (*Mii and Taira, 2009*). Multiple comparisons were carried out with pairwise Wilcoxon rank sum test (two-sided) in which significance levels (*p*-values) were adjusted by the Holm method, using R.

## Mathematical modeling

Two ligand components were considered: free and bound. This model includes five dynamic processes: (i) production of ligands, (ii) diffusion of the free component in intercellular space, (iii) binding of ligands to dotted structures ('docking sites') such as HS clusters on the surface of cells, (iv) release of the bound component from 'docking sites,' and (v) internalization of the bound component into cells. The model in one-dimensional space is written as:

$$\frac{\partial u}{\partial t} = D\frac{\partial^2 u}{\partial x^2} - a(x)u + bv + g(x), (0 < x < L) \tag{3}$$

$$\frac{dv}{dt} = a(x)u - bv - cv, (0 < x < L) \tag{4}$$

where *u* and *v* represent the amounts of free and bound components of morphogen molecules, respectively. The symbols *a(x)*, *b*, *c*, and *g(x)* represent binding, release, internalization, and

production rates, respectively. $D$ represents the diffusion coefficient of the free component in the extracellular space. The ligand is assumed to be produced in a limited region using the following function:

$$g(x) = \begin{cases} g_{max} & (0 \leq x \leq R) \\ 0 & (\text{elsewhere}) \end{cases}. \tag{6}$$

We assumed that the binding rate $a(x)$ depends on the position $x$, following heterogeneous distribution of HS clusters on the cell surface. The following function was used for $a(x)$:

$$a(x) = \begin{cases} a_{n,max} & (p_1 n \leq x \leq P_1 n + p_2, n = 1, 2, ...) \\ 0 & (\text{elsewhere}) \end{cases}, \tag{7}$$

where $p_1$ and $p_2$ are the interval and width, respectively, of docking sites. We used the no-flux (Neumann) boundary condition at $x = 0$ and $L$. We calculated the model by numerical simulation. The initial distributions of $u$ and $v$ were set at 0 throughout the entire space. In the one-dimensional space, distributions of free ($u$) and bound ($v$) components of secreted proteins were obtained by computer simulation, where the spatial length $L = 1000$ (μm). Distributions of $u$ (red) and $v$ (blue) are presented at time $t = 100$ (sec), which almost reached steady states. We used the forward difference method with the spatial step $\Delta x = 0.1$ and temporal step $\Delta t = 0.0001$ in numerical calculations. In *Figure 5B*, parameter values are: $D = 20.0$, $a_{n,max} = 10.0$, $b = 0.1$, $c = 0.1$, $g_{max} = 0.2$, $R = L/1000$, $p_1 = 2$, and $p_2 = 0.2$. In other panels, distributions in specific conditions are shown (see figure legends).

## Acknowledgements

We thank Dr. Min Kyun Park for hybridoma culture and 5'RACE for IgG cDNAs. We also thank Dr. A Miyawaki for Venus cDNA; Dr. R Moon for pCS2+Xwnt8; Dr. M Affolter for morphotrap cDNA; Dr. W Hashimoto for the HepIII plasmid (pET21b-Phep_3797); Dr. Shinya Matsuda for critical reading; Drs. Hiroshi Koyama, Yohei Kondo and Motosuke Tsutsumi for discussion and Dr. Steven D Aird for editing and proofreading. We thank the Confocal Microscope core facility at the ConveRgence mEDIcine research cenTer (CREDIT), Asan Medical Center. This work was supported in part by following programs: KAKENHI (24870031, 15K14532, 18K14720 to YM; 19H05670 to AM; 24657147, 18H02447, 25251026 to MT; 17K19418, 18H02454, 19H04797 to ST), the NINS program for cross-disciplinary study (1311608, 01311801 to YM), Joint Usage/Research Center program of Institute for Frontier Life and Medical Sciences Kyoto University, JST-PRESTO (JPMJPR194B to YM), JST-CREST (JPMJCR13W6, JPMJCR1922 to AM).

## Additional information

### Funding

| Funder | Grant reference number | Author |
| --- | --- | --- |
| Japan Science and Technology Agency | JPMJPR194B | Yusuke Mii |
| Japan Society for the Promotion of Science | 24870031 | Yusuke Mii |
| Japan Society for the Promotion of Science | 15K14532 | Yusuke Mii |
| Japan Society for the Promotion of Science | 18K14720 | Yusuke Mii |
| National Institutes of Natural Sciences | 1311608 | Yusuke Mii |
| National Institutes of Natural Sciences | 01311801 | Yusuke Mii |
| Japan Society for the Promotion of Science | 24657147 | Yusuke Mii Masanori Taira |

| | | |
|---|---|---|
| Japan Society for the Promotion of Science | 18H02447 | Masanori Taira |
| Japan Society for the Promotion of Science | 25251026 | Masanori Taira |
| Japan Science and Technology Agency | JPMJCR13W6 | Atsushi Mochizuki |
| Japan Science and Technology Agency | JPMJCR1922 | Atsushi Mochizuki |
| Japan Society for the Promotion of Science | 19H05670 | Atsushi Mochizuki |
| Japan Society for the Promotion of Science | 17K19418 | Shinji Takada |
| Japan Society for the Promotion of Science | 18H02454 | Shinji Takada |
| Japan Society for the Promotion of Science | 19H04797 | Shinji Takada |

The funders had no role in study design, data collection and interpretation, or the decision to submit the work for publication.

## Author contributions

Yusuke Mii, Conceptualization, Data curation, Formal analysis, Funding acquisition, Validation, Investigation, Visualization, Methodology, Writing - original draft, Writing - review and editing; Kenichi Nakazato, Software, Formal analysis, Investigation; Chan-Gi Pack, Data curation, Formal analysis, Methodology; Takafumi Ikeda, Resources, Investigation; Yasushi Sako, Investigation, Methodology; Atsushi Mochizuki, Software, Supervision, Writing - original draft; Masanori Taira, Shinji Takada, Conceptualization, Investigation, Supervision, Funding acquisition, Writing - original draft, Writing - review and editing

## Author ORCIDs

Yusuke Mii (iD) https://orcid.org/0000-0002-1907-5665
Chan-Gi Pack (iD) http://orcid.org/0000-0002-6578-3099
Shinji Takada (iD) https://orcid.org/0000-0003-4125-6056

## Ethics

Animal experimentation: All experiments using *Xenopus laevis* were approved by the Institutional Animal Care and Use Committee, National Institutes of Natural Sciences (Permit Number 18A038, 19A062, 20A053), or the Office for Life Science Research Ethics and Safety, University of Tokyo.

## Decision letter and Author response

Decision letter https://doi.org/10.7554/eLife.55108.sa1
Author response https://doi.org/10.7554/eLife.55108.sa2

# Additional files

## Supplementary files

- Source code 1. Source code for *Figure 5B* and *Figure 5—figure supplement 1D*.
- Source code 2. Source code for *Figure 5C*.
- Source code 3. Source code for *Figure 5D*.
- Source code 4. Source code for *Figure 5E*.
- Source code 5. Source code for *Figure 5F*.
- Source code 6. Source code for *Figure 5—figure supplement 1A*.

• Source code 7. Source code for *Figure 5—figure supplement 1E*. All source code files are written in C. An executable file 'a.out' will be generated by compilation. By executing 'a.out', amounts of $u$ (free) and $v$ (bound) at each position in the field is recorded in a data file 'dists_uv.dat'.

• Supplementary file 1. Primers used for molecular cloning of IgG cDNAs from hybridomas. See the section of 'cDNA cloning of IgG from cultured hybridomas' in Materials and methods for details.

• Transparent reporting form

### Data availability

Sequence data for anti-HA IgG genes have been deposited in Genbank/DDBJ under accession codes LC522514 and LC522515.

The following datasets were generated:

| Author(s) | Year | Dataset title | Dataset URL | Database and Identifier |
|---|---|---|---|---|
| Mii Y | 2020 | *Mus musculus* mRNA for immunoglobulin gamma 2b, complete cds | https://www.ncbi.nlm.nih.gov/nuccore/LC522514 | NCBI GenBank, LC522514.1 |
| Mii Y | 2020 | *Mus musculus* mRNA for immunoglobulin kappa, complete cds | https://www.ncbi.nlm.nih.gov/nuccore/LC522515 | NCBI GenBank, LC522515.1 |

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

## Appendix 1

### Detecting freely diffusing molecules with confocal microscopy

When we consider detection of diffusing molecules with confocal microscopy, diffusing molecules might not be detected because of their large displacement. In our imaging conditions, pixel dwell time was 24.4 µs and observed maximum $D_{fast}$ in FCS was about 80 µm$^2$/s (Figure S3E). During pixel dwell time $t$, mean square displacement (MSD) of molecules of diffusion coefficient $D$ in three-dimensional space is $6Dt$ (*Crank, 1975*). Therefore, maximum MSD is estimated as

$$6 \times 80 \times 24.4 \times 10^{-6} = 0.0117 \mu m^2$$

Accordingly, mean displacement of the molecules at the maximum is

$$\sqrt{\mathrm{MSD}} = 0.108 \mu m = 108 nm$$

On the other hand, when the numerical aperture (*NA*) of an objective lens is given, the Reyligh diffraction limit in our conditions is calculated as

$$\frac{0.61\lambda}{NA} = \frac{0.61 \times 500}{1.1} = 277 \mathrm{nm}$$

This value of the diffraction limit is larger than the $\sqrt{\mathrm{MSD}}$ so that even the most rapidly diffusing molecules observed in FCS measurements, loss of fluorescence due to diffusion is not likely to occur. Indeed, we detected slight but significant increase of photon counts with sec-mV (*Figure 1D* and *Figure 1—figure supplement 2*) with a long pixel dwell time. We speculate that the difficulty of visualizing sec-mV is mainly due to narrowness of extracellular space in *Xenopus* embryos

