## [Decision Letter]

**Acceptance summary:**

How secreted signaling proteins disperse to form gradients in living animals is a fundamental question for developmental and cell biology. Mii et al., use quantitative light microscopy to analyze dispersal dynamics of a tagged Wnt and FrzB in *Xenopus* embryos. The authors provide evidence that the spatial distribution of binding sites and dynamic exchange of bound and freely diffusing proteins facilitate rapid formation of long-range protein gradients and that a similar mechanism could apply to other pathways as well.

**Decision letter after peer review:**

Thank you for submitting your article "Quantitative analyses reveal extracellular dynamics of Wnt ligands in *Xenopus* embryos" for consideration by *eLife*. Your article has been reviewed by 3 peer reviewers, one of whom is a member of our Board of Reviewing Editors, and the evaluation has been overseen by Naama Barkai as the Senior Editor. The reviewers have opted to remain anonymous.

The reviewers have discussed the reviews with one another and the Reviewing Editor has drafted this decision to help you prepare a revised submission.

As the editors have judged that your manuscript is of interest, but as described below that additional experiments are required before it is published, we would like to draw your attention to changes in our revision policy that we have made in response to COVID-19 (https://elifesciences.org/articles/57162). First, because many researchers have temporarily lost access to the labs, we will give authors as much time as they need to submit revised manuscripts. We are also offering, if you choose, to post the manuscript to bioRxiv (if it is not already there) along with this decision letter and a formal designation that the manuscript is 'in revision at *eLife*'. Please let us know if you would like to pursue this option. (If your work is more suitable for medRxiv, you will need to post the preprint yourself, as the mechanisms for us to do so are still in development.)

Summary:

In this manuscript, Mii et al., use quantitative light microscopy to analyze dispersal dynamics of a tagged Wnt and FrzB in *Xenopus* embryos. Using protein trapping approaches, they demonstrate that Venus-tagged Wnt8 and FrzB can spread across many cell diameters in vivo and that high levels of secreted protein can be captured at a distance from their sources. Importantly, Mii et al., showed using FCS that only a small fraction of the detectable protein was freely diffusing at a given time while the majority was not. Mathematical models of Wnt8 and FrzB dispersal based on experimentally derived parameters including exchange between the free and bound populations were also able to recapitulate a gradient profile for bound extracellular proteins. The authors propose that the spatial distribution of ligand binding sites and on-off dynamics facilitate formation of long-range Wnt protein gradients, and that a similar mechanism could apply to other pathways.

How signaling proteins disperse in living animals is a fundamental question, and a variety of mechanisms are likely to operate in different contexts. This manuscript presents a series of targeted experiments that provide an important step forward for the field. However, there are key issues that need to be addressed to give confidence in the biological relevance of the results and the mathematical techniques.

Essential revisions:

Comments from the three reviewers have been condensed into a single list to help with revisions. Successfully addressing the reviewers' essential revision points 1-3 will require additional laboratory experiments, while the others likely will not. Successfully addressing the reviewers' points 11-14 would normally require new experiments, but based on current *eLife* guidelines due to COVID-19, I think that these could instead be adequately addressed in the manuscript text if the additional experiments are not practical at this point.

1. All three reviewers had similar reservations regarding the use of overexpression for quantitative analyses. Most importantly, this paper relies entirely on overexpressed Wnt ligand. As the proposed mechanisms controlling Wnt diffusion involve binding to extracellular factors, altering the stoichiometry of Wnt ligands and binding sites may affect the normal balance between the different pools of Wnt. This would be especially problematic if overexpression saturates endogenous mechanisms that limit Wnt diffusion within the areas used for measurements. Similar concerns also apply to the FrzB measurements. To ensure confidence in the physiological relevance of the data, the authors should perform the key experiments over a range of lower mRNA dosages down to the minimum required for FCS to test if the measured parameters (diffusion constants, fraction of molecules in each pool) are the same. If the results differ, would it be possible to extrapolate to a case of zero overexpression?

2. The authors should attempt to estimate how the levels of the tagged proteins compare to the endogenous ones. For Wnt8, this could potentially be done by comparing immunofluorescence levels for the endogenous and the tagged Wnt8.

3. Functionality of tagged proteins.

a. Please provide additional data to demonstrate that the tagged proteins used here are fully functional. This revision is considered critical due to the level of precision in the analyses. Showing only that Wnt or FrzB fusion proteins can induce a phenotype when overexpressed would not be sufficient. The authors could use TOP-FLASH assays like in Figure 4 Supplement 1 over a range of lower concentrations to confirm that the responses at low mRNA levels are the same for tagged and untagged Wnt8 and FrzB.

b. Based on Figure 4, supplement 1, the mKikGR-FrzB fusion protein is not fully functional and should be removed.

4. Some of the details of the proposed mechanism are confusing. The paper seems to measure both slow and fast diffusing components from the FCS measurements and also to propose an immobile fraction that cannot be measured in FCS. Or do the authors believe that the slow component in FCS can be equated with the immobile one? If the former, do the authors really need three different pools of ligand to explain the data? On the other hand, the model only has two components a fast diffusing one and an immobile fraction – would the results be affected by adding a slow diffusing component as measured in FCS? or alternatively treating the immobile fraction as slowly diffusing?

5. In light of (4), the authors should consider whether simpler models fit their data equally well. For example, the effective diffusion model in Figure 4 supplement 3 that has fewer parameters appears to fit the Frzb data better than the two population model in the main text. For Wnt, it isn't clear whether goodness of fit is improved by adding a second component as more parameters are also added.

6. Can the authors speculate about why the sec-mv has a slow diffusing component? This seems surprising. Again, the authors should justify using a two-population (slow and fast) rather than a simpler model.

7. There may be an error in the FDAP equation in figure 4. The data appear to be normalized so that f(0) = 1, however, plugging 0 into the equation does not give 1. More likely, the authors intend to fit to the function f(t) = (1-c)*exp(-koff*t)+c which has the correct limiting behavior at both t = 0 and t = inf. This shows however that there are only two parameters to be fit (this makes sense, they correspond to the decay rate and the immobile fraction) so it is unclear how the authors are fitting three parameters from their data. Essentially, one cannot extract k_on_ from this data – it is lumped into the prefactor of the exponential.

8. Modeling

a. The argument in lines 402-411 is not valid, as can be seen with numerical simulations, in which a molecule with D=0.05 um^2/s does reach the end of a 200 μm long patterning field over 24 h and can be readily absorbed by a localized sink there.

b. In fact, the models (effective diffusion or binding/dissociation) are not mutually exclusive (as described in the Crank reference mentioned by the authors). It would therefore be useful for the readers if the authors could present a unified description of restricted, effective and hindered diffusion.

c. Since the proteins are detected over distances of ~15 cell diameters (~200 um), the spatial range of the simulations should be increased to the length of a *Xenopus* embryo to avoid reflection from the boundary. It would also be ideal if the authors could measure the gradient throughout the embryo with FCS (similar to PMID 19741606) or note their limitations.

d. Please normalize each curve to the maximum of each species to see if there really is a difference in the ranges of the free and bound components in the model.

e. The authors state that "Notably, in our mathematical model, distributions of both free and bound components converged to steady states within a few minutes, showing rather fast dynamics in the context of embryonic patterning". The cause for these unexpectedly fast dynamics might be the extremely high off-rates compared to previously measured values (e.g. compare to the values used in PMID 22445299). Depending on re-assessment of the FDAP data, the authors should repeat their simulations. It would be ideal to more directly compare the simulations to the experimental data (e.g. overlay data from Figure 2D with simulations) and to compare the fast kinetics prediction of the model to the temporal gradient evolution in real embryos.

f. Please simulate the outcome of FDAP experiments (i.e. the dissipation of concentration from one peak). Is an increase in neighboring regions detected, and how does this compare with the experimental findings?

g. Please simulate the Sec-mV findings. Maybe the simulations will clarify why this protein is not visible?

h. Please clarify why the data in Figure 5F looks different from panel B, even though it should be the same according to the figure legend.

9. FCS analyses and inference of bound/unbound fractions

a. The authors state that "[…] diffusion coefficients measured with FRAP and FCS differ by 3-4 orders of magnitude (Rogers and Schier, 2011)", but this statement is outdated and should be corrected (see e.g. PMIDs 23533171 and 28919007).

b. Figure 3B shows problematic drift that might underlie the long correlation times.

c. The authors further state that "By autocorrelation analysis, FCS can measure the diffusion coefficient (D) and the number of particles, which is equivalent to the concentration of the diffusing molecules (Figure 3E), but it cannot measure immobile molecules (Hess et al., 2002)", but this statement (and similar ones in lines 220-221 and 237-238) is imprecise and needs to be corrected. Avalanche photodiodes or the like used in FCS experiments do detect all signal emitted from fluorescent molecules, but the inference of diffusion coefficients depends on mobile molecules.

d. These imprecisions in FCS theory lead to unsuitable analyses in Figure 3G. Importantly, not only the inferred number of particles but also the count rate is higher for Sec-mV, although the same mRNA amount was used for all constructs. The difference in the outcomes between confocal imaging and FCS might instead be the result of variability between experiments. The authors should therefore measure and compare the intensities using photon counting and FCS within the same embryos if possible or note their limitations.

e. The subsequent considerations about immobile and diffusing molecules (e.g. lines 230-233) are inappropriate and should be corrected. The current logic is: 9.4 detected mV-Wnt8 particles constitute 43% of the 21.7 particles detected for sec-mV, and since 43% of the 19.5 maximally detectable mobile mean photon counts equal 8.4 mean photon counts, the rest of the mean photon counts (219.7) for mV-Wnt8 is immobile; however, this logic is inappropriate since identical amounts of mRNA are not equimolar given the protein size differences, identical mRNA amounts do not necessarily give rise to similar protein amounts due to differences in translation or stability, and day-to-day or instrument differences might influence the intensities.

f. Please use the Akaike information criterion or the like to compare the 2-component fits with more parsimonious 1-component fits of the FCS data, especially given the small differences in the highly and poorly mobile fractions.

g. Please mention from how many independent embryos the FCS measurements were derived.

10. In Figure 3 supplement 1, the comparison is not appropriate given the very different sample numbers and the small effect size. The authors should instead use bootstrapping approaches with similar sample numbers.

11. For the FDAP experiments, the instruments may just not have the sensitivity to detect small numbers of photoconverted molecules that spread locally, especially considering that a large amount of fluorescence loss is due to photobleaching and that the imaging is at a single plane. The authors should take care with interpreting these data unless they are able to repeat the experiments using new acquisition parameters to minimize bleaching. Additionally, as the authors noted that the pseudo-equilibrium binding constant may be underestimated because curve fitting does not consider photobleaching, and the simulations crucially depend on these parameters, they should execute new simulations with the parameters determined in the absence of photobleaching if possible. If repeating these experiments is not practical due to current restrictions the authors should temper their conclusions.

12. A goal of this manuscript is to link quantitative measurements of local protein dynamics to larger spatiotemporal patterns of extracellular protein dispersal in embryos. To do so, the authors assume that short-term measurements at sub-apical junctions fully capture the processes that are important for protein dispersal and gradient formation. The hypothesis is that local dynamics measured by FCS and FDAP in a particular location can be extrapolated to model mechanisms for long distance dispersal across fields of many cells, but this has not been shown experimentally. The authors could use FRAP experiments photobleaching a large area at a distance from the Wnt source followed by time-lapse imaging over a long period to confirm that the pattern and timing of fluorescence recovery across many cells is consistent with predictions from their model. Under normal circumstances, these experiments would be considered required revisions. However, in light of the current research situation it would be acceptable to instead discuss these limitations in the manuscript.

13. The manuscript seems to treat Wnt8 and FrzB dispersal as occurring within a single plane, with the dynamics at sub-apical junctions reflecting dynamics elsewhere. However, the tagged proteins could be spreading along the basal surfaces or through entirely different routes or mechanisms. These possibilities should be considered in the discussion if it is not feasible to assess them experimentally.

14. Similarly to (12), a key assumption is that the freely diffusing population of Wnt detectable by FCS is the protein population that moves between cells, but this cannot be directly concluded from the experiments. The authors should discuss the possibility of Wnt transport on cytonemes/signaling filopodia or contacts between cells at earlier developmental stages. Based on the *Xenopus* fate map, can the authors rule out potential mechanisms for Wnt gradient formation based on cell lineages or migration patterns? The morphotrap experiments argue against these possibilities, but the fact that tagged protein can accumulate to high levels in distant trap-expressing cells could be explained by higher stability of trapped versus untrapped protein.

[Editors' note: further revisions were suggested prior to acceptance, as described below.]

Thank you for submitting your article "Quantitative analyses reveal extracellular dynamics of Wnt ligands in *Xenopus* embryos" for consideration by *eLife*. Your revised article has been evaluated by a Reviewing Editor, one Reviewer, and Naama Barkai as the Senior Editor.

The Reviewing Editor has drafted this to help you prepare a revised submission.

The authors have satisfactorily addressed the reviewers' combined points, and the additional experiments have substantially strengthened the conclusions. I am pleased to recommend accepting this paper for publication pending minor revisions. There are only three points to address prior to full acceptance.

Summary:

How secreted signaling proteins disperse to form gradients in living animals is a fundamental question for developmental and cell biology. Mii et al., use quantitative light microscopy to analyze dispersal dynamics of a tagged Wnt and FrzB in *Xenopus* embryos. The authors provide evidence that the spatial distribution of binding sites and dynamic exchange of bound and freely diffusing proteins facilitate rapid formation of long-range protein gradients and that a similar mechanism could apply to other pathways as well.

Essential Revisions:

1. The authors state "We provide the source code for our mathematical model, written in C.", but we could not find the code among the current manuscript items. Please provide this code as a Supplementary file.

2. The diffusion coefficients for 250 pg and 20 pg of mV-Wnt8 appear to differ between Figure 3D and Figure 3—figure supplement 1B, and the authors should check whether the correct data are plotted.

3. The figure legend for Figure 4 includes a panel F, but this image is missing from the figure. Please provide an updated Figure 4 including F or remove it from the legend.

4. The authors may wish to note in the text that reduced activity of mV-tagged Wnt8 compared to untagged Wnt8 could possibly be due, at least in part, to differences in translation.

---

## [Author Response]

Essential revisions:Comments from the three reviewers have been condensed into a single list to help with revisions. Successfully addressing the reviewers' essential revision points 1-3 will require additional laboratory experiments, while the others likely will not. Successfully addressing the reviewers' points 11-14 would normally require new experiments, but based on current eLife guidelines due to COVID-19, I think that these could instead be adequately addressed in the manuscript text if the additional experiments are not practical at this point.

Thank you very much for constructive and helpful comments to improve our manuscript. Accordingly, we performed essential experiments addressing the reviewers’ points, mainly estimation of endogenous-equivalent doses of mV-Wnt8 and -Frzb, their activity, and FCS analyses with endogenous-level expression. In addition, we performed FCS analyses with HS digestion using newly developed, membrane-tethered Heparinase III. In the current situation, it is difficult for us to perform the FDAP analyses again. However, we carefully re-assessed our FDAP data, and revised our manuscript, in accordance with the reviewers’ suggestions. We hope that our revisions have adequately addressed their concerns.

1. All three reviewers had similar reservations regarding the use of overexpression for quantitative analyses. Most importantly, this paper relies entirely on overexpressed Wnt ligand. As the proposed mechanisms controlling Wnt diffusion involve binding to extracellular factors, altering the stoichiometry of Wnt ligands and binding sites may affect the normal balance between the different pools of Wnt. This would be especially problematic if overexpression saturates endogenous mechanisms that limit Wnt diffusion within the areas used for measurements. Similar concerns also apply to the FrzB measurements. To ensure confidence in the physiological relevance of the data, the authors should perform the key experiments over a range of lower mRNA dosages down to the minimum required for FCS to test if the measured parameters (diffusion constants, fraction of molecules in each pool) are the same. If the results differ, would it be possible to extrapolate to a case of zero overexpression?2. The authors should attempt to estimate how the levels of the tagged proteins compare to the endogenous ones. For Wnt8, this could potentially be done by comparing immunofluorescence levels for the endogenous and the tagged Wnt8.

Thank you for these suggestions. We performed immunostaining of overexpressed mV-Wnt8 in the animal cap region with rabbit antiserum against Wnt8 (Mii et al., 2017) to determine a dose equivalent to endogenous distribution of Wnt8 in the ventral marginal zone (VMZ, just over the ventral mesoderm expressing *wnt8*, and highest for Wnt8 distribution). The data show that injection of 20 pg/embryo of mV-Wnt8 mRNA provides Wnt8 staining in the animal cap region that is equivalent to that in the VMZ (Figure 1—figure supplement 2A). Endogenous expression levels of *Xenopus* embryonic mRNAs were published in Session et al., *Nature* 2016 (PMID 27762356). In this data, *wnt8a* (*L* + *S*) shows 288 transcripts per million (TPM) and *frzb* (*L* + *S*) shows 180 TPM. Thus, we estimated an endogenous-equivalent dose of *mV-frzb* as 10-20 pg/embryo.

In response to comment 1, we performed FCS measurements at the endogenous-equivalent level (by injecting 20 pg/embryo mRNA) of mV-Wnt8 and mV-Frzb (new Figure 3 and Figure 3—figure supplement 1). Previous FCS measurements were performed with a Zeiss system at RIKEN (Wako, Saitama); however, due to the pandemic, it is currently difficult to perform experiments at RIKEN. Instead, we performed new measurements with a Leica system at our institute (Okazaki, Aichi). New data with an endogenous-equivalent level of mV-Wnt8 and mV-Frzb also showed a freely diffusing component as well as a slowly diffusing component (Figure 3D and Figure 3—figure supplement 1B).

3. Functionality of tagged proteins.a. Please provide additional data to demonstrate that the tagged proteins used here are fully functional. This revision is considered critical due to the level of precision in the analyses. Showing only that Wnt or FrzB fusion proteins can induce a phenotype when overexpressed would not be sufficient. The authors could use TOP-FLASH assays like in Figure 4 Supplement 1 over a range of lower concentrations to confirm that the responses at low mRNA levels are the same for tagged and untagged Wnt8 and FrzB.b. Based on Figure 4, supplement 1, the mKikGR-FrzB fusion protein is not fully functional and should be removed.

In accordance with this comment, we performed TOP-FLASH assays at an endogenous-equivalent dose (20 pg/embryo) for mV-Wnt8 and -Frzb, in conditions such that the activity of the reporter is not saturated (Figure 1—figure supplement 2 B and C). The data indicate that these tagged proteins possess activities to activate (Wnt8) or to inhibit (Frzb) canonical Wnt signaling. mV-Frzb showed similar signaling activity to the wild type. But mV-Wnt8 showed 1/3 to 1/2 of the full activity. However, we want to continue using this tagged protein for this study for the following reason.

In our previous studies, we used mEGFP-Wnt3a to investigate biochemical and biophysical features of Wnt3a (Takada, Mii et al., *Commun. Biol.* 2018, PMID 30320232). In this study, mEGFP-Wnt3a showed roughly one-fourth the activity of non-tagged Wnt3a, as assayed with endogenous ß-catenin in L cells (Supplementary Figure 1; Because L cells do not express Cadherins, we can measure canonical Wnt activity of conditioned media by determining the amounts of ß-catenin in L cells). The activity of mEGFP-Wnt3a might look insufficient; however, we demonstrated that homozygous mEGFP-knock-in mice that express the same fusion protein from endogenous *Wnt3a* loci are viable and show normal development (Shinozuka et al., *Development*, 2019, PMID 30651295). Furthermore, mEGFP-knock-in mice show similar distributions of mEGFP-Wnt3a and a signaling range (assayed with *Axin2* expression) comparable to that of wild-type mice. mV-Wnt8 used in this study has a similar tagging design, and shows similar spikes in FCS measurements (Figure 3B), suggesting formation of multimeric homo-complexes as shown in Takada et al., (2018). In addition, no haploinsufficiency has been reported for *wnt* and *sfrp* genes as far as we know. Taking these facts into consideration, we think that use of mV-Wnt8 and mK-Frzb is justified.

4. Some of the details of the proposed mechanism are confusing. The paper seems to measure both slow and fast diffusing components from the FCS measurements and also to propose an immobile fraction that cannot be measured in FCS. Or do the authors believe that the slow component in FCS can be equated with the immobile one? If the former, do the authors really need three different pools of ligand to explain the data? On the other hand, the model only has two components a fast diffusing one and an immobile fraction – would the results be affected by adding a slow diffusing component as measured in FCS? or alternatively treating the immobile fraction as slowly diffusing?

We apologize that details of FCS data and our model were confusing. As we mentioned in response to comment 9f, a 2-component model, considering fast and slow components in FCS is supported by AIC. Comparison of photon count imaging and FCS, and the FDAP experiment suggest a slower or immobile population in addition to the two components in FCS. Therefore, experimental evidence supports at least three different populations. On the other hand, in mathematical modeling, we mainly consider an immobile and a freely diffusing population for simplicity. We also examined modeling with an immobile and a slowly diffusing (*D* = 0.50 um^2^/s) population, but the range of the distribution was very short (Figure 5—figure supplement 1A). We made these relationships clear in the text (lines 551-553).

5. In light of (4), the authors should consider whether simpler models fit their data equally well. For example, the effective diffusion model in Figure 4 supplement 3 that has fewer parameters appears to fit the Frzb data better than the two population model in the main text. For Wnt, it isn't clear whether goodness of fit is improved by adding a second component as more parameters are also added.

In the revised version, we used a simpler model for fitting FDAP data (f(t) = (1-c)*exp(-koff*t)+c, suggested in comment 7). The goodness of fit of this model was similar to that of the effective diffusion model, as expected (both consider 2 parameters).

6. Can the authors speculate about why the sec-mv has a slow diffusing component? This seems surprising. Again, the authors should justify using a two-population (slow and fast) rather than a simpler model.

Thank you for these comments. We also think that this is an interesting point. As described in our answer about AIC in FCS fitting (reviewer comment 9f), newly obtained sec-mV data also show slow components, which are supported by AIC. To characterize slow components, we performed two-types of experiments. One is a cross-correlation analysis of mV-Wnt8 and a membrane-tracer (Lyn-miRFP703), and the other is enzymatic digestion of heparan sulfate (HS). In cross-correlation measurements, no cross-correlation was observed between mV-Wnt8 and lyn-miRFP703, suggesting that even the slow component is not associated with cell membranes (Figure3—figure supplement 1C, D).

For HS digestion experiments, we used a newly constructed membrane-tethered heparinase III (HepIII-HA-GPI), which enables us to compare HS-digested regions and control regions in the same embryos, to sec-mV and mV-Wnt8. In sec-mV, a slight, but statistically significant increase (about 5%) of the ratio of the fast components was observed, without a significant change in the number of particles (Figure 3 G, H). We speculate that this might reflect an increase of free space in the intercellular space, because HS is a highly charged polymer, so it is highly space-filling when hydrated. On the other hand, the number of particles, as well as the ratio of the fast components of mV-Wnt8 were significantly increased by HS-digestion (Figure 3E, F), probably reflecting the release of mV-Wnt8 from specific binding to HS chains.

Considering that there are many other types of cell surface and extracellular matrix molecules in addition to HS, we speculate that the slow components may reflect the narrow extracellular space of *Xenopus* ectoderm, occupied with those extracellular molecules. This could be understood as a kind of “hindered diffusion”, but the spatial scale is significantly smaller than the previously proposed idea at a multicellular level (Figure 3—figure supplement 1E, Muller et al., 2013).

7. There may be an error in the FDAP equation in figure 4. The data appear to be normalized so that f(0) = 1, however, plugging 0 into the equation does not give 1. More likely, the authors intend to fit to the function f(t) = (1-c)*exp(-koff*t)+c which has the correct limiting behavior at both t = 0 and t = inf. This shows however that there are only two parameters to be fit (this makes sense, they correspond to the decay rate and the immobile fraction) so it is unclear how the authors are fitting three parameters from their data. Essentially, one cannot extract k_on_ from this data – it is lumped into the prefactor of the exponential.

Thank you for pointing this out. We derived the FDAP equation as follows, starting from the case of reaction dominant, equation (11) in Sprague et al., *Biophys. J.* (2004) (PMID 15189848).frap(t)=1−κon∗κon∗+κoff∗exp⁡(−κofft) (11)

This equation is to fit a complete recovery (to 1); however, experimental data for FRAP often show incomplete recovery (to *I_final_*); thus, we derivedfrap(t)=Ifinal{1−κon∗κon∗+κoff∗exp⁡(−κofft)} (11′).

As *fdap*(*t*) *=* 1 *– frap*(*t*),fdap(t)=1−Ifinal{1−κon∗κon∗+κoff∗exp⁡(−κofft)}=1−Ifinal+Ifinalκon∗κon∗+κoff∗exp⁡(−κofft)=(1−C)κon∗κon∗+κoff∗exp⁡(−κofft)+C(oldequation1),where *C* = 1 – *I_final_*. In eq. (11), *frap*(0) does not give 1 as,frap(0)=1−κon∗κon∗+κoff∗=κoff∗κon∗+κoff∗=feq.

This is because, in the reaction dominant case, also known as “diffusion-uncoupled” FRAP, the concentration of freely diffusing molecules rapidly converges to *f_eq_*. Thus, eq. (11) and also its derivative, eq. (1) do not give 1 at *t* = 0.

Nonetheless, after considering reviewer comments, we are currently unsure whether our way of adjusting for the incomplete recovery (eq. 11') and fitting such a “delicate” model to our data containing non-negligible bleach are appropriate, especially for extracting *k^*^_on_*. So we decided to use the suggested model, *fdap*(*t*) = (1-*c*)*exp(-*k_off_***t*)+*c*. Our data contain an effect of photobleaching, but this can be corrected with fixed mKikGR-Frzb data. Using this model and bleach correction with the chemically-fixed samples, we derived new *k_off_* values (Figure 4D, E, Figure 4—figure supplement 2D).

8. Modelinga. The argument in lines 402-411 is not valid, as can be seen with numerical simulations, in which a molecule with D=0.05 um^2/s does reach the end of a 200 μm long patterning field over 24 h and can be readily absorbed by a localized sink there.

We agree, so we deleted the argument.

b. In fact, the models (effective diffusion or binding/dissociation) are not mutually exclusive (as described in the Crank reference mentioned by the authors). It would therefore be useful for the readers if the authors could present a unified description of restricted, effective and hindered diffusion.

Thank you for this suggestion. We added discussion about these models in lines 776-784.

c. Since the proteins are detected over distances of ~15 cell diameters (~200 um), the spatial range of the simulations should be increased to the length of a *Xenopus* embryo to avoid reflection from the boundary. It would also be ideal if the authors could measure the gradient throughout the embryo with FCS (similar to PMID 19741606) or note their limitations.

In accordance with this suggestion, we changed the length of the field of modeling from 100 μm to 1000 um. The revised version of Figure 5 shows results from the 1000 μm field; however, the increase of the field length did not affect the distributions of the ligands. Therefore, distributions within 100 μm from the source are presented, as in the previous versions. This is also reasonable because Wnt-based AP-patterning in early *Xenopus* embryos may occur in/around the dorsal lip region of the gastrula, which is far smaller than the whole embryo.

FCS measurements require mounting embryos in LMP agarose to avoid movements of embryos; thus, it is technically difficult to measure gradients in a wide field. We noted this technical difficulty in lines 707-708.

d. Please normalize each curve to the maximum of each species to see if there really is a difference in the ranges of the free and bound components in the model.

Following this suggestion, we prepared normalized curves for free and bound species to compare their ranges for Figure 5, B and C (Figure 5—figure supplement 1, B and C). The *l* values obtained by fitting with *C*(*x*) *= C_o_*exp(*– x/l*) were slightly different between the free and bound species (*lu* and *lv*, respectively); however, we can see that the free and bound species show similar ranges.

e. The authors state that "Notably, in our mathematical model, distributions of both free and bound components converged to steady states within a few minutes, showing rather fast dynamics in the context of embryonic patterning". The cause for these unexpectedly fast dynamics might be the extremely high off-rates compared to previously measured values (e.g. compare to the values used in PMID 22445299). Depending on re-assessment of the FDAP data, the authors should repeat their simulations. It would be ideal to more directly compare the simulations to the experimental data (e.g. overlay data from Figure 2D with simulations) and to compare the fast kinetics prediction of the model to the temporal gradient evolution in real embryos.

Thank you for this suggestion. In our re-assessment of FDAP data, we obtained *k_off_* values of mV-Wnt8 and mV-Frzb as 0.16 and 0.58 (s-1), respectively. These values are much higher than the values used in Zhou et al., 2012 (10^-7^ s-1); however, they correspond to Dpp-Receptor kinetics. In the same paper, the authors mentioned that “binding of BMPs to glypicans (and heparan sulfate in general) is reversible, with modest affinity and relatively rapid kinetics.” In our system, HS appears to be a major scaffold of Wnt8 and Frzb; thus, large *k_off_* values may be reasonable. In another report (Muller et al., Science 2012), the clearance rate constants, *k_1_*, which are equivalent to *k_off_*, are around 10^-4^ s-1; however, in their experiments, the area of photoconversion was considerably larger, containing tens of cells. In this condition, re-binding or exchanging reactions of photoconverted molecules should contribute to the small k1 values. Regarding our observation, simulation of FDAP experiments recapitulate the decline of photoconverted molecules in a single peak in 16 seconds (Figure 5—figure supplement 1E, see also the next answer).

f. Please simulate the outcome of FDAP experiments (i.e. the dissipation of concentration from one peak). Is an increase in neighboring regions detected, and how does this compare with the experimental findings?

Following this comment, we simulated the FDAP experiment (Figure 5—figure supplement 1E, simulating Figure-4 Figure supplement 2). Unlike the experiment with mK-Wnt8, modeling shows a small increase in neighboring regions (but this is similar to the case of mK-Frzb). For this difference, we mainly consider two possibilities; i) our imaging system may not be sufficiently sensitive to detect such a small increase, or the increase of ligand may be cancelled by photobleaching. ii) if binding dynamics of mK-Wnt8 are slower than that in our model, ligands may diffuse away before re-binding in neighboring regions.

g. Please simulate the Sec-mV findings. Maybe the simulations will clarify why this protein is not visible?

We think that modeling itself would not clarify why sec-mV is not visible (in our model, sec-mV is not degraded; thus, it will just increase continuously). Freely diffusing molecules, such as cytosolic GFP, at a similar concentration (~ 100 uM) is readily visible. We speculate that invisibility of sec-mV is mainly due to the narrowness of the extracellular space in *Xenopus* embryos.

h. Please clarify why the data in Figure 5F looks different from panel B, even though it should be the same according to the figure legend.

We apologize for this mistake. Actually, Figure 5B in the previous version was presented incorrectly (probably a result with different parameters). We corrected Figure 5B. The *l* value in previous version was calculated with the correct data. As mentioned above, figures in Figure 5 were replaced with new modeling using the wider field (1000 um), but the result is almost indistinguishable from the previous version. In addition, we changed the style of the previous Figure 5—figure supplement 1 (new Figure supplement 1C), with normalization of the plots, similar to other panels.

9. FCS analyses and inference of bound/unbound fractions

We removed “concentration” from tables of FCS data to avoid misleading data, after considering that the width of intercellular space (< 30 nm) of *Xenopus* embryos appears narrower than a diameter of confocal detection volume (~ 200-300 nm). Also, we checked data in old Figure 3F again, and corrected the table (new Figure 3—figure supplement 2D).

a. The authors state that "[…] diffusion coefficients measured with FRAP and FCS differ by 3-4 orders of magnitude (Rogers and Schier, 2011)", but this statement is outdated and should be corrected (see e.g. PMIDs 23533171 and 28919007).

Thank you for pointing this out. We deleted this part. Instead, we mentioned that their “optimal ranges for diffusion coefficients differ” (lines 128-129).

b. Figure 3B shows problematic drift that might underlie the long correlation times.

In general, such drift could affect correlation time. But in the case of Figure 3B, the drift is much slower than diffusion-related movements, thus it is negligible. To avoid unfavorable drift, we set the duration of a single measurement to 5 seconds (previous measurements) or 10 seconds (new measurements for this revision). To avoid misunderstanding, only traces of single measurements are presented in our new figures.

c. The authors further state that "By autocorrelation analysis, FCS can measure the diffusion coefficient (D) and the number of particles, which is equivalent to the concentration of the diffusing molecules (Figure 3E), but it cannot measure immobile molecules (Hess et al., 2002)", but this statement (and similar ones in lines 220-221 and 237-238) is imprecise and needs to be corrected. Avalanche photodiodes or the like used in FCS experiments do detect all signal emitted from fluorescent molecules, but the inference of diffusion coefficients depends on mobile molecules.

We apologize for this imprecise statement. We corrected this part (lines 244-247).

d. These imprecisions in FCS theory lead to unsuitable analyses in Figure 3G. Importantly, not only the inferred number of particles but also the count rate is higher for Sec-mV, although the same mRNA amount was used for all constructs. The difference in the outcomes between confocal imaging and FCS might instead be the result of variability between experiments. The authors should therefore measure and compare the intensities using photon counting and FCS within the same embryos if possible or note their limitations.e. The subsequent considerations about immobile and diffusing molecules (e.g. lines 230-233) are inappropriate and should be corrected. The current logic is: 9.4 detected mV-Wnt8 particles constitute 43% of the 21.7 particles detected for sec-mV, and since 43% of the 19.5 maximally detectable mobile mean photon counts equal 8.4 mean photon counts, the rest of the mean photon counts (219.7) for mV-Wnt8 is immobile; however, this logic is inappropriate since identical amounts of mRNA are not equimolar given the protein size differences, identical mRNA amounts do not necessarily give rise to similar protein amounts due to differences in translation or stability, and day-to-day or instrument differences might influence the intensities.

Photon counting and FCS within the same embryo is technically difficult. After considering the questions raised by reviewers regarding our estimation (Figure 3G), we removed it. On the other hand, we see a large difference in photon counts of mV-Wnt8 and sec-mV, whereas *NoP* measured by FCS of mV-Wnt8 and sec-mV were rather similar. We speculate that FCS measurements might be biased to choose positions where HS-bound molecules are not abundant. Otherwise HS-bound, immobile molecules cause photobleaching, which results in fluorescence intensity drift.

f. Please use the Akaike information criterion or the like to compare the 2-component fits with more parsimonious 1-component fits of the FCS data, especially given the small differences in the highly and poorly mobile fractions.

Thank you for this suggestion. We examined AIC to compare 1- and 2-component models with new measurements according to a previous report (Tsutsumi et al., 2016) (Figure 3—figure supplement 1A. Evaluation of previous data with AIC is currently difficult for us, mainly because we cannot access the facility to perform the analysis). For all averaged data analyzed, the 2-component model was better than the 1-component model, judging by AIC (Figure 3—figure supplement 1A). For individual measurements (10 seconds), most measurements fit the 2-component model better than the 1-component model. Thus, we concluded that the 2-component model is suitable to fit our FCS data from *Xenopus* embryos.

g. Please mention from how many independent embryos the FCS measurements were derived.

We are sorry for our failure to have supplied this information. We added numbers of embryos measured for FCS in the legend or tables.

10. In Figure 3 supplement 1, the comparison is not appropriate given the very different sample numbers and the small effect size. The authors should instead use bootstrapping approaches with similar sample numbers.

Thank you for this suggestion. For old Figure 3 supplement 1 (new Figure 3—figure supplement 2), we removed the statistical test, after considering its requirement. Instead, we used bootstrapping for examining significance in Figure 3—figure supplement 1B. Using R software, we did bootstrapping as follows.

#NoP (mV-Wnt8 250 pg vs sec-mV)

> library(boot)

> N=10000

> data_join <- c(nw25, ns)

> theta <- numeric(N)

> for(i in 1:N){

data1_boot <- sample(data_join, 20, replace = TRUE)

data2_boot <- sample(data_join, 26, replace = TRUE)

theta[i] <- mean(data1_boot)-mean(data2_boot)

}

> dif <- mean(nw25) - mean(ns)

> sum(theta >= dif)/N

[1] 2567 #not significant

11. For the FDAP experiments, the instruments may just not have the sensitivity to detect small numbers of photoconverted molecules that spread locally, especially considering that a large amount of fluorescence loss is due to photobleaching and that the imaging is at a single plane. The authors should take care with interpreting these data unless they are able to repeat the experiments using new acquisition parameters to minimize bleaching. Additionally, as the authors noted that the pseudo-equilibrium binding constant may be underestimated because curve fitting does not consider photobleaching, and the simulations crucially depend on these parameters, they should execute new simulations with the parameters determined in the absence of photobleaching if possible. If repeating these experiments is not practical due to current restrictions the authors should temper their conclusions.

We sincerely considered this comment. Currently, it is very difficult for us to perform FDAP experiments because of the COVID-19 pandemic, and also because we currently do not have access to the confocal microscope suitable for FDAP. As we noted in our response to comment 7, we used a simpler model and photobleaching was corrected with chemically-fixed samples.

12. A goal of this manuscript is to link quantitative measurements of local protein dynamics to larger spatiotemporal patterns of extracellular protein dispersal in embryos. To do so, the authors assume that short-term measurements at sub-apical junctions fully capture the processes that are important for protein dispersal and gradient formation. The hypothesis is that local dynamics measured by FCS and FDAP in a particular location can be extrapolated to model mechanisms for long distance dispersal across fields of many cells, but this has not been shown experimentally. The authors could use FRAP experiments photobleaching a large area at a distance from the Wnt source followed by time-lapse imaging over a long period to confirm that the pattern and timing of fluorescence recovery across many cells is consistent with predictions from their model. Under normal circumstances, these experiments would be considered required revisions. However, in light of the current research situation it would be acceptable to instead discuss these limitations in the manuscript.

Thank you for this comment. We noted these limitations in the Discussion (lines 895-899).

13. The manuscript seems to treat Wnt8 and FrzB dispersal as occurring within a single plane, with the dynamics at sub-apical junctions reflecting dynamics elsewhere. However, the tagged proteins could be spreading along the basal surfaces or through entirely different routes or mechanisms. These possibilities should be considered in the discussion if it is not feasible to assess them experimentally.

Thank you for this comment. Judging from our data, we can see accumulation of mV-Wnt8 and mV-Frzb (low dose) in the proximal region (to the source) in the morphotrap-expressing area (Figure 2C, D), consistent with the idea of dispersal in a single plane, at least in the animal cap region. Of course, in actual embryogenesis, a diffusing molecule could diffuse three-dimensionally in embryonic tissue, we discussed these possibilities in the Discussion (lines 886-891). However, aim of our model is rather to provide a basic framework to understand our observation of protein distributions and quantitative analyses than to construct a complex model considering the three-dimensional shape of *Xenopus* embryos.

14. Similarly to (12), a key assumption is that the freely diffusing population of Wnt detectable by FCS is the protein population that moves between cells, but this cannot be directly concluded from the experiments. The authors should discuss the possibility of Wnt transport on cytonemes/signaling filopodia or contacts between cells at earlier developmental stages. Based on the *Xenopus* fate map, can the authors rule out potential mechanisms for Wnt gradient formation based on cell lineages or migration patterns? The morphotrap experiments argue against these possibilities, but the fact that tagged protein can accumulate to high levels in distant trap-expressing cells could be explained by higher stability of trapped versus untrapped protein.

Thank you for this suggestion. Because the superficial layer of *Xenopus* embryos, including the animal cap region, is a kind of epithelium, in which cells are tightly packed, cell-movement-based transfer of these proteins can be excluded together with distributions of source and morphotrap-expressing cells. We have never observed cytoneme/filopodia-based transfer of Wnt protein in the superficial layer, but these protrusions could be involved in mesenchymal cells, which are packed more loosely. We discussed these possibilities in the Discussion (lines 891-895). We agree with the possibility of higher stability of trapped protein. We think that both dispersal and stabilization of protein are required for accumulation on the morphotrap-expressing cells.

[Editors' note: further revisions were suggested prior to acceptance, as described below.]

Essential Revisions:1. The authors state "We provide the source code for our mathematical model, written in C.", but we could not find the code among the current manuscript items. Please provide this code as a Supplementary file.

We apologize for the lack of source codes. Please find the uploaded source codes.

2. The diffusion coefficients for 250 pg and 20 pg of mV-Wnt8 appear to differ between Figure 3D and Figure 3—figure supplement 1B, and the authors should check whether the correct data are plotted.

We are grateful that the reviewers pointed this out. We suppose they suggest that mean values shown in Figure 3D appear to differ from values indicated with bold horizontal lines in Figure 3—figure supplement 1B. We confirmed that mean values of diffusion coefficients of mV-Wnt8 (250 or 20 pg) shown in Figure 3D and plots of *D_fast_* and *D_slow_* shown in Figure 3—figure supplement 1B are based on the same data. We would like to note that the bold horizontal lines in Figure Supplement 1B indicate median values (50 percentile), which generally differ from mean values in asymmetric distributions. To avoid misunderstanding, we added “Mean values are presented.” to the legend of Figure 3D.

3. The figure legend for Figure 4 includes a panel F, but this image is missing from the figure. Please provide an updated Figure 4 including F or remove it from the legend.

We apologize for this mistake. In the revised version, the legend for panel F has been removed.

4. The authors may wish to note in the text that reduced activity of mV-tagged Wnt8 compared to untagged Wnt8 could possibly be due, at least in part, to differences in translation.

We appreciate this suggestion. We added this point to the text.